# Single-atom nanozymes catalytically surpassing naturally occurring enzymes as sustained stitching for brain trauma

Shaofang Zhang[1,6], Yonghui Li[1,6], Si Sun[1,6], Ling Liu[2], Xiaoyu Mu [2], Shuhu Liu[3], Menglu Jiao[1], Xinzhu Chen[2], Ke Chen[2], Huizhen Ma[1], Tuo Li[4], Xiaoyu Liu[1], Hao Wang[2], Jianning Zhang[4], Jiang Yang [5] & Xiao-Dong Zhang [1,2] ✉

Regenerable nanozymes with high catalytic stability and sustainability are promising substitutes for naturally-occurring enzymes but are limited by insufficient and non-selective catalytic activities. Herein, we developed single-atom nanozymes of $RhN_4$, $VN_4$, and $Fe-Cu-N_6$ with catalytic activities surpassing natural enzymes. Notably, $Rh/VN_4$ preferably forms an $Rh/V-O-N_4$ active center to decrease reaction energy barriers and mediates a "two-sided oxygen-linked" reaction path, showing 4 and 5-fold higher affinities in peroxidase-like activity than the $FeN_4$ and natural horseradish peroxidase. Furthermore, $RhN_4$ presents a 20-fold improved affinity in the catalase-like activity compared to the natural catalase; $Fe-Cu-N_6$ displays selectivity towards the superoxide dismutase-like activity; $VN_4$ favors a 7-fold higher glutathione peroxidase-like activity than the natural glutathione peroxidase. Bioactive sutures with $Rh/VN_4$ show recyclable catalytic features without apparent decay in 1 month and accelerate the scalp healing from brain trauma by promoting the vascular endothelial growth factor, regulating the immune cells like macrophages, and diminishing inflammation.

Artificial enzyme engineering with sustainabilities emerges as a versatile methodology for creating biocatalysts[1–6]. The intrinsic feature of continuous electron transfers allows artificial enzymes to maintain high catalytic stability and arbitrarily tailor their characteristics[7–12]. The superior catalytic activity over the naturally-occurring enzymes is one of the major challenges existing in designing artificial enzymes[13–16]. A feasible strategy is to construct an artificial active center by imitating the biologically active center of enzymes or proteins[2,17–22]. As a fast growing category of artificial enzymes, the emerging nanozyme can actively tailor biocatalytic activities and selectivity via its flexible atomic structures and molecular engineering[23–26]. State-of-art nanozymes with an $M-N_4$ center inspired by cytochrome P450 have superior reaction rates and high substrate affinities close to the natural horseradish peroxidase (HRP)[27–29]. Expanded $Fe-N_5$ and $Fe-N_3-P$ were reported to have 17- and 2-fold higher catalytic efficiencies than $Fe-N_4$[14,30], indicating their great potential in boosting bioactivities. However, the development of nanozymes with properties outperforming natural enzymes remains unfulfilled and highly challenging.

Other unresolved and high-profile questions are the underlying reaction mechanisms and detailed atomic coordination structures

[1]Department of Physics and Tianjin Key Laboratory of Low Dimensional Materials Physics and Preparing Technology, School of Sciences, Tianjin University, Tianjin 300350, China. [2]Tianjin Key Laboratory of Brain Science and Neural Engineering, Academy of Medical Engineering and Translational Medicine, Tianjin University, Tianjin 300072, China. [3]Beijing Synchrotron Radiation Facility (BSRF), Institute of High Energy Physics (IHEP), Chinese Academy of Sciences (CAS), Beijing 100049, China. [4]Department of Neurosurgery and Key Laboratory of Post-trauma Neuro-repair and Regeneration in Central Nervous System, Tianjin Medical University General Hospital, Tianjin 300052, China. [5]State Key Laboratory of Oncology in South China, Collaborative Innovation Center for Cancer Medicine, Sun Yat-sen University Cancer Center, Guangzhou 510060, China. [6]These authors contributed equally: Shaofang Zhang, Yonghui Li, and Si Sun. ✉e-mail: xiaodongzhang@tju.edu.cn

during biocatalytic reactions to endow catalytic selectivity[13,31–33]. Unlike the well-defined structures of single-atom catalysts[16,34,35], traditional nanostructures without elucidated atomic structures on the surface cannot provide sufficient structure-function correlation to study the electron transfers at atomic levels[36]. For example, Pt and $Fe_3O_4$ exhibit peroxidase-like (POD-like) activities[6,37]; Au and $V_2O_5$ are more selective for the glutathione peroxidase-like (GPx-like) activity[38–40]; Cu demonstrates a superior superoxide dismutase-like (SOD-like) activity[41]. However, the selectivity of these biocatalysts merely relies on the specific atomic coordination and surface ligands of individual nanosystems, making it difficult to corroborate a universal catalytic mechanism explicitly. Besides, the complexity of reaction pathways also compromises catalytic reactions of nanozymes[33], leaving the atomic configuration and bond morphology inadequately clarified during the dynamic reaction process.

In this work, we report structurally well-defined and atomically-precise biocatalysts of $RhN_4$ and $VN_4$ with high POD- and catalase (CAT)-like activities, surpassing corresponding natural enzymes and reported $FeN_4$[42–44]. Moreover, the $RhN_4$ and $VN_4$ favorably form the Rh/V-O-$N_4$ structure as the active center of circular catalytic processes to decrease reaction energy barriers significantly, which is fundamentally different from $FeN_4$[14,45–47] (Fig. 1). In particular, the catalytic mechanism of $RhN_4$ and $VN_4$ modulates a unique "two-sided oxygen-linked" catalytic reaction path, resulting in high catalytic efficiency. Meanwhile, the specific SOD activity is derived from Fe-Cu-$N_6$, and the GPx-like activity of $VN_4$ considerably exceeds that of the natural GPx. Furthermore, the high catalytic stability and recyclability enable these nanozymes to be used as medical sutures for sustained scalp healing from brain trauma via a serial biocatalytic process. $RhN_4$ and $VN_4$ in sutures play an essential role in the proactive regulation and immune control of macrophages and vascular endothelial growth factors (VEGF).

## Results

### Structural characterization of $MN_x$

The $MN_x$ (M = Rh, V, Fe, Cu, and Fe-Cu; $x$ = 4 or 6) nanozymes were synthesized via pyrolysis[14,48,49]. $RhN_4$ and $VN_4$ preserved the pristine polyhedral structure of zinc-imidazole frameworks (ZIF-8) as visualized by high-resolution transmission electron microscopy (HRTEM), while Fe-Cu-$N_6$ exhibited an amorphous morphology with abundant micropores, similar to $FeN_4$[50] (Fig. 2a, d, g). In addition, aberration-corrected high-angle annular dark-field scanning transmission

electron microscopic (AC-HAADF-STEM) images showed single metal atoms sparsely dispersed in the ZIP-8 framework (Fig. 2b, e, h). The Fe-Cu diatomic pairs could be observed in the enlarged magnified AC-HAADF-STEM image of Fe-Cu-$N_6$ (Supplementary Fig. 1a). The average distances between Fe and Cu atoms were calculated as ~2.27 Å from the statistical intensity profiles (Supplementary Fig. 1b). For energy-dispersive X-ray spectroscopy (EDS) mapping analysis, an individual polyhedron was selected. Single metal atoms and C and N elements are uniformly distributed throughout the entire nanostructure (Fig. 2c, f, i, and Supplementary Fig. 2). Characteristic peaks of metal oxides and atomic crystallites were not detected in the ultraviolet-visible (UV-vis) absorption spectra, X-ray diffraction (XRD), and Raman spectra, indicating the absence of metal nanoparticles (Supplementary Figs. 3–5). X-ray photoelectron spectroscopy (XPS) identified major nitrogen species as pyridinic, graphitic, and pyrrolic N, while carbon species were primarily composed of graphitic C, C=C-N, and N-C=N (Supplementary Figs. 6–8). The elemental composition of metal-based active sites was investigated by inductively coupled plasma-mass spectrometry (ICP-MS). The mass percentages of Rh and V in Rh/$VN_4$ nanozymes were only 0.19 and 0.22 %, while Fe, Cu, and Fe-Cu in $FeN_4$, $CuN_4$, Fe-Cu-$N_6$ nanozymes were elevated at 1.4, 1.39, and 1.1 %, respectively (Supplementary Table 1).

Next, we characterized the atomic coordination structure between metal atoms and neighboring atoms by the X-ray absorption near-edge structure (XANES), along with the Fourier transformed (FT) extended X-ray absorption fine structure (EXAFS). By analyzing the XANES of $RhN_4$ (Fig. 2j), we can observe that the Rh species become more positively charged than Rh(acac)$_3$, which may originate from the Rh-$N_4$ unit. Furthermore, $VN_4$ showed a characteristic peak at ~1.58 Å, shorter than the bulk V-V bond of ~2.29 Å, suggesting the dispersed state of individual V atoms (Fig. 2k and Supplementary Fig. 9). Similarly, the major peaks of Fe and Cu atoms for Fe-Cu-$N_6$, $FeN_4$, and $CuN_4$ were located at ~1.47 and ~1.48 Å (Fig. 2l, Supplementary Figs. 10–13, and Supplementary Table 2), corresponding to Fe-N and Cu-N bonds, respectively[51–55]. To characterize the exact Fe-Cu coordination, we thermodynamically and kinetically predicted five possible structures of Fe-Cu-$N_x$ based on the above findings, defined as Fe-Cu-$N_6$, Fe-Cu-$N_8$-I, Fe-Cu-$N_8$-II, Fe-Cu-$N_8$-III, and Fe-Cu-$N_8$-IV (Supplementary Fig. 14). The simulated distance of Fe-Cu-$N_6$ was 2.46 Å (Supplementary Table 3), which closely matched the experimental result of 2.27 Å. Fe-Cu-$N_6$ was identified as the most thermodynamically favorable

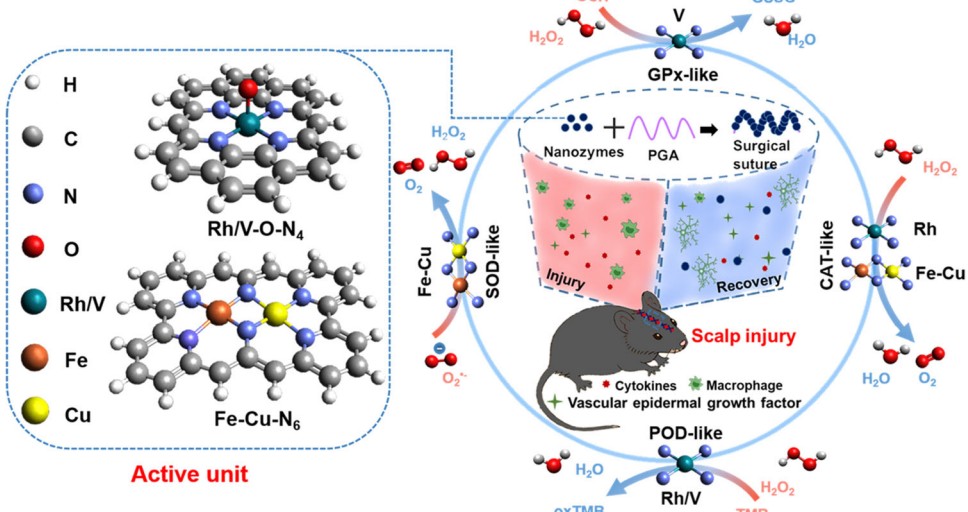

**Fig. 1 | Single-atomic nanozymes with superior catalytic activities and selectivity beyond the natural enzymes were developed with unique definitive units.** The bioactive sutures based on artificial enzymes showed catalytic recyclability and sustainability without evident decay in activities. They also accelerated the scalp healing of brain trauma via promoting the vascular endothelial growth factor, and decreasing oxidative stress and inflammation.

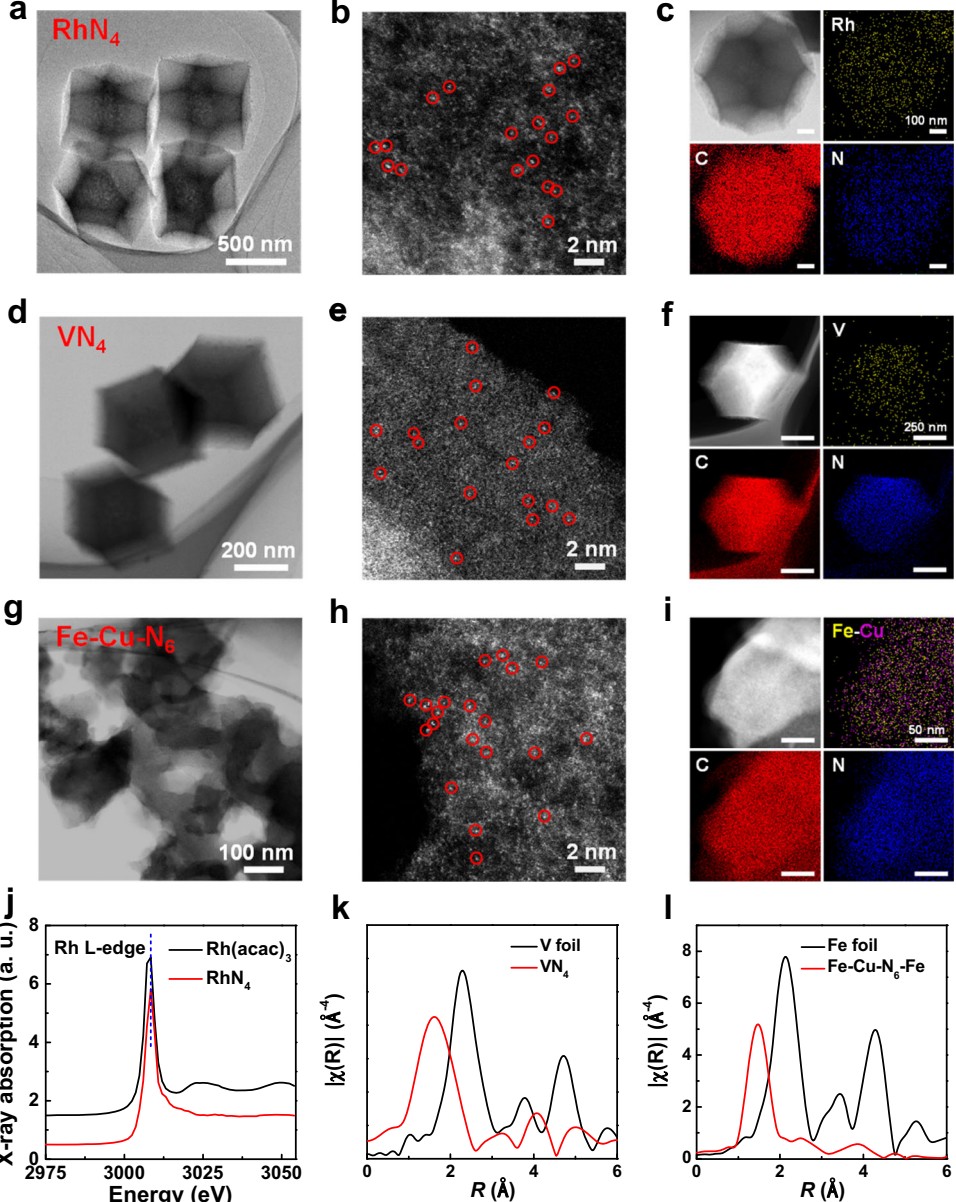

**Fig. 2 | Structural characterization of MN$_x$.** TEM, AC-HAADF-STEM, and corresponding EDS mapping images of **a**–**c** RhN$_4$, **d**–**f** VN$_4$, and **g**–**i** Fe-Cu-N$_6$, respectively ($n$ = 3 images from three independent samples). Single metal atoms highlighted by red circles. **j** XANES spectra of RhN$_4$ and the Rh(acac)$_3$ reference at the rhodium L-edge. **k, l** Fourier-transformed magnitude of the V K-edge EXAFS signal of VN$_4$ and Fe K-edge EXAFS signal of Fe-Cu-N$_6$, along with the respective bulk metal references. Fourier transforms were not corrected for phase shifts. R represents the radial distribution function. 'a. u.' represents arbitrary units.

structure due to the lowest formation energy of −5.432 eV by the Density Functional Theory (DFT) simulation (Supplementary Table 3). Therefore, all of the subsequent calculations of the enzyme-like pathways were based on the Fe-Cu-N$_6$ structure.

**POD-like catalytic activities of MN$_x$**

We assessed the POD-like activities of MN$_x$ by monitoring the time-lapse reaction rates at equivalent concentrations using the 3,3',5,5'-tetramethylbenzidine (TMB) colorimetric assay (Fig. 3a), which were correlated well with substrate concentrations (Supplementary Figs. 15, 16). The specific activities of MN$_x$ were quantified by referring to the activity curve of HRP standards. RhN$_4$ and VN$_4$ were 4.37 and 2.22 U/μmol in POD-like activities, about 8 times higher than the widely reported FeN$_4$ (0.514 U/μmol) and CuN$_4$ (0.286 U/μmol)[47] (Fig. 3b). In addition, we continuously monitored the enzymatic activity for 1 month. All MN$_x$ displayed outstanding long-term stability with

negligible decays, while the natural HRP rapidly lost its activity within 2 days (Fig. 3c). Besides, significant intrinsic kinetic differences between MN$_x$ were observed, including their Michaelis–Menten constants ($K_m$) and catalytic constants ($k_{cat}$). The catalytic reaction affinities of RhN$_4$ ($K_m \sim 55.7$ μM) and VN$_4$ ($K_m \sim 65.6$ μM) to the TMB substrate were ~4–5 times better than those of FeN$_4$ ($K_m \sim 184.6$ μM) and the natural HRP ($K_m \sim 276.2$ μM), denoting their excellent catalytic characteristics (Supplementary Table 4, Supplementary Figs. 17, 18). The kinetic parameters of MN$_x$ for H$_2$O$_2$ substrates were also quantified (Supplementary Figs. 19, 20, and Supplementary Table 5). Compared to the reported $K_m$ values of single-atom catalysts[42,44,47,56], MN$_x$ showed a superior binding affinity for H$_2$O$_2$, confirming the outstanding catalytic properties. However, all single-atom nanozymes exhibited lower binding affinities for H$_2$O$_2$ substrates than HRP, which remains an unconquered challenge that would need further improvements.

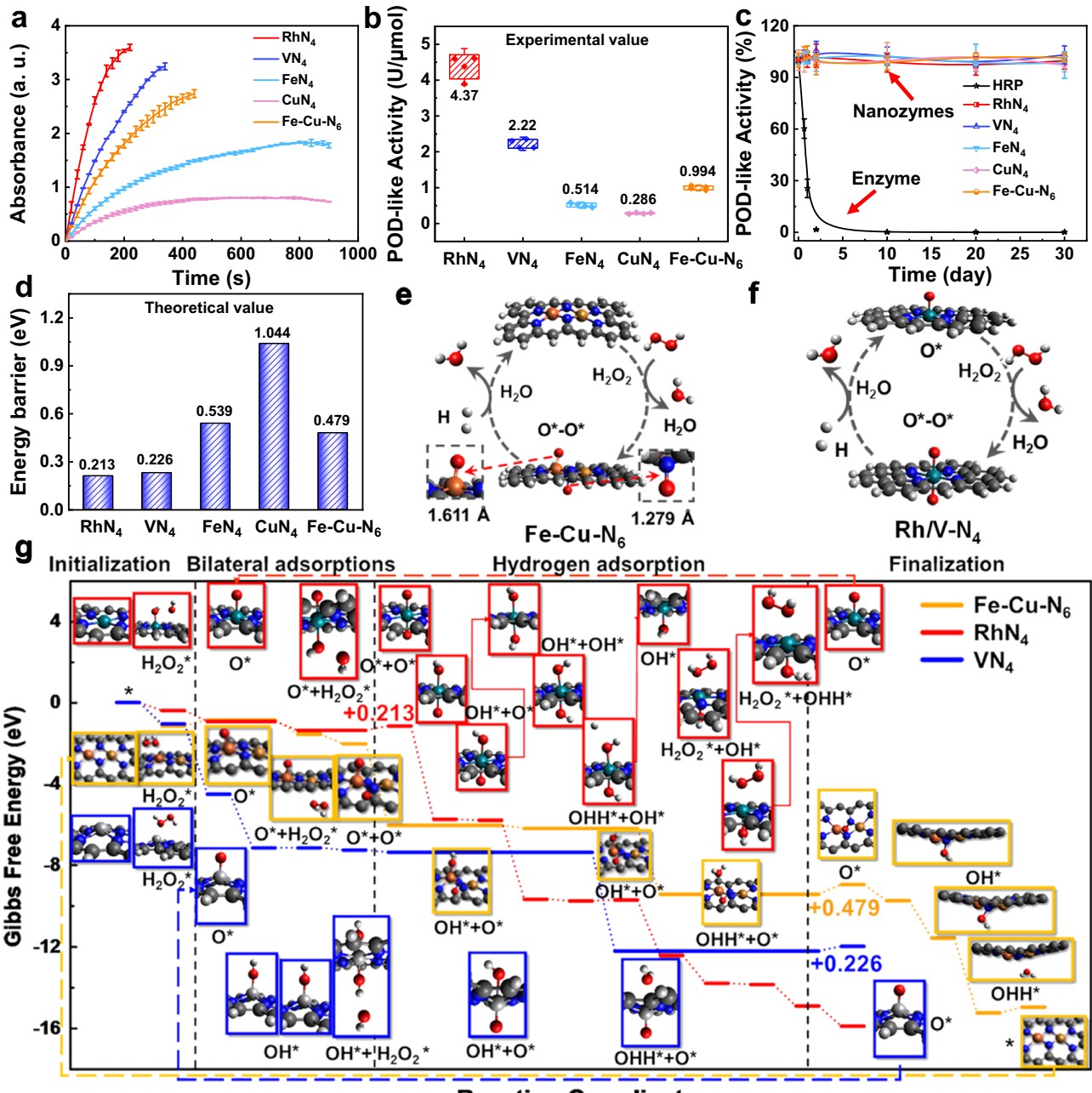

**Fig. 3 | The POD-like activity of MN$_x$. a** Reaction-time curves of the TMB colorimetric reaction catalyzed by 1 μM MN$_x$, with the substrate concentration of TMB and H$_2$O$_2$ at 800 μM and 0.2 M ($n$ = 3 independent experiments). 'a. u.' represents arbitrary units. **b** Quantification of specific POD-like activities of MN$_x$ ($n$ = 4 independent experiments; boxes represent the median and interquartile range (IQR) and the upper and lower whiskers extending to the values that are within 1.5 × IQR). The specific activity value (U/μmol) was determined by dividing the POD-like activity by the metal-based active sites. **c** Comparison of the stability of POD-like activities between MN$_x$ and natural enzymes ($n$ = 3 independent experiments). **d** The energy barrier of MN$_x$ in POD-mimic reaction pathway via DFT simulation. **e**, **f** Schematic illustration of the cyclable catalytic POD processes for **e** Fe-Cu-N$_6$ and **f** Rh/VN$_4$. **g** POD processes with MN$_x$ (Rh, V, and Fe-Cu). White, dark gray, blue, red, cyan, light gray, orange, and brown balls represent H, C, N, O, Rh, V, Fe, and Cu atoms. The MN$_x$ catalyst is represented by an asterisk (*) for clarity. All tests are performed at room temperature. All data are presented as mean ± SD.

To explore the underlying reaction pathways and substantiate the high POD-like catalytic efficiency, we investigated a variety of intermediate/transition states of MN$_x$ attached to different chemical units using DFT. The simulated catalytic pathways are summarized in Fig. 3g and Supplementary Fig. 21. By comparing the three simulated reaction pathways, we derived four serial steps: initialization, bilateral adsorptions, hydrogen adsorption, and finalization, which are present in all catalysts except for CuN$_4$. The catalytic process begins with the adsorptions of H$_2$O$_2$ molecules onto the active sites of metal catalysts.

Next, each transition state yields an "initialized" intermediate state by releasing a water molecule, with an extra oxygen atom adsorbed at the activation site. Such an "initialized" state takes advantage of the material's planar structural geometry, thus making further absorption on the other side feasible. Finally, in the bilateral adsorption step, another H$_2$O$_2$ molecule can be captured by the active sites from the opposite side of the "initialized" state. Such an "initialized" transient state is key to the high catalytic efficiency: the stably attached oxygen reduces the ionization of the metallic atom, therefore inducing decomposition of the

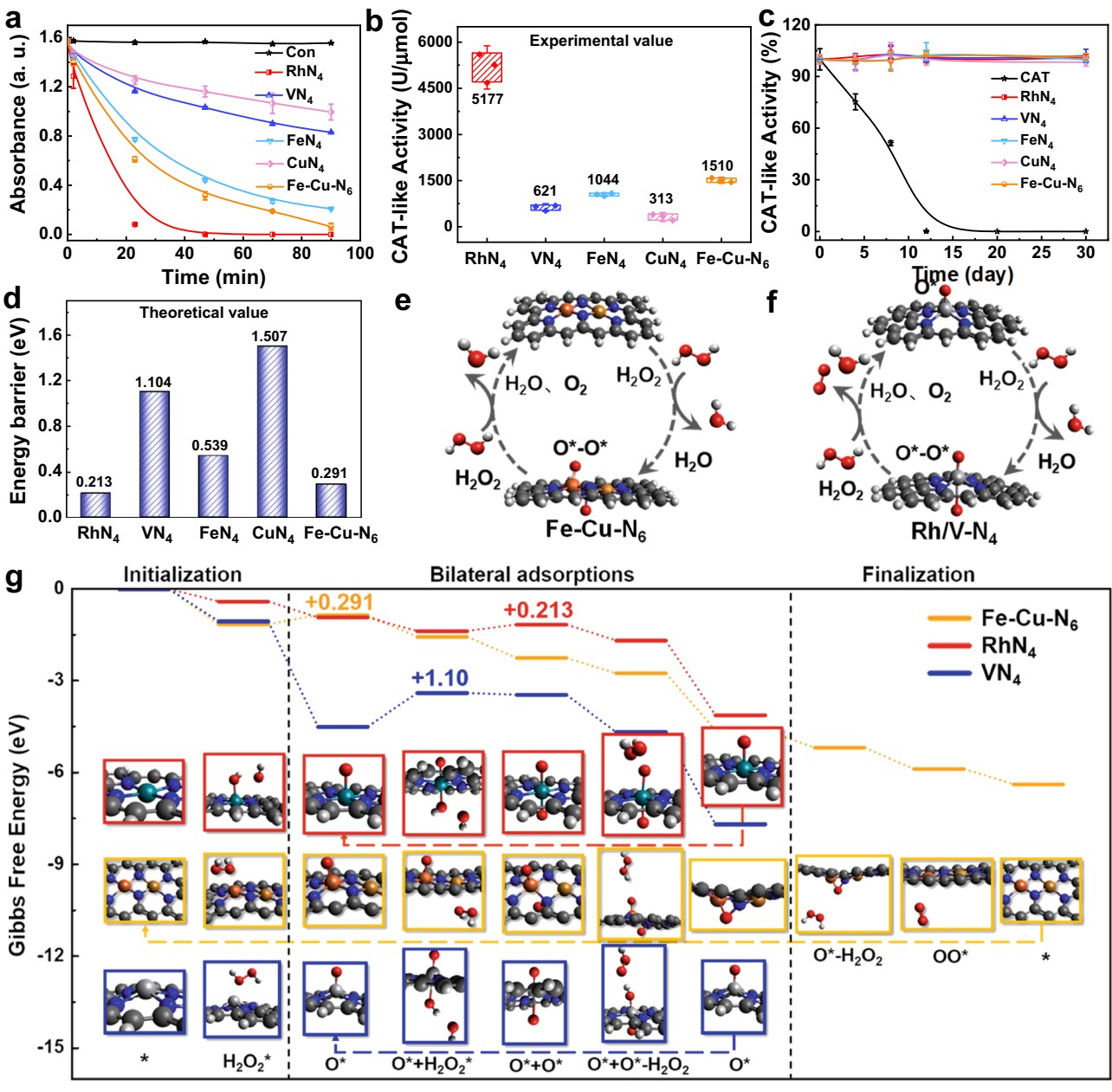

**Fig. 4 | The CAT-like activity of MN$_x$. a** Reaction-time curves of the decomposition of H$_2$O$_2$ catalyzed by 4.8 μM MN$_x$ ($n = 3$ independent experiments). 'a. u.' represents arbitrary units. **b** Quantification of specific CAT-like activities of MN$_x$. One enzyme activity unit represents the amount of MN$_x$ that catalyzes the decomposition of 1 μmol H$_2$O$_2$ within 1 minute ($n = 3$ independent experiments; boxes represent the median and IQR and the upper and lower whiskers extending to the values that are within 1.5 × IQR). **c** Comparison of the stability of CAT-like activities between MN$_x$ and natural enzymes ($n = 3$ independent experiments). **d** The energy barriers of MN$_x$ in the CAT-mimic reaction simulated by DFT. **e, f** Schematic illustration of the cyclable catalytic CAT processes for MN$_x$. **e** Fe-Cu-N$_6$ and **f** Rh/VN$_4$. **g** The CAT processes with MN$_x$ (Rh, V, and Fe-Cu). Dark gray, blue, red, cyan, light gray, orange, and brown balls represent H, C, N, O, Rh, V, Fe, and Cu atoms. The MN$_x$ catalyst is represented by an asterisk (*) for clarity. **e** The energy barriers of MN$_x$ in the CAT-mimic reaction simulated by DFT. All tests were performed at room temperature. All data are presented as mean ± SD. Some error bars are too small to be visible.

H$_2$O$_2$ molecule more efficiently. We calculated the associated activation energy in each reaction pathway, defined as the highest energy barrier, to evaluate the catalytic efficiency qualitatively (Fig. 3d). The predicted activation energies of RhN$_4$ (0.213 eV) and VN$_4$ (0.226 eV) suggest their high efficiency of catalysis, agreeing with the experimental results. The rate-determining steps (RDSs) associated with the activation energies in the reaction pathways are: the reinstatement process to release the water molecule for VN$_4$, the beginning of the finalization step for Fe-Cu-N$_6$, and the adsorption of the first hydrogen atom for RhN$_4$. In the dual center system of Fe-Cu-N$_6$ with two activation sites, the bilateral adsorption shows a significant difference in bond lengths of the attached oxygen atoms (Fig. 3e). Surprisingly, CuN$_4$ does not include the "bilateral adsorption" process due to the incapability to extract another oxygen from the H$_2$O$_2$ molecule on the other side (the energy barrier is 1.84 eV). With oxygen atoms attached on both sides, hydrogen adsorptions occur to complete the formation of water molecules. In addition, there are no direct correlations between the bilateral adsorption and hydrogen adsorption steps. The results reveal that the

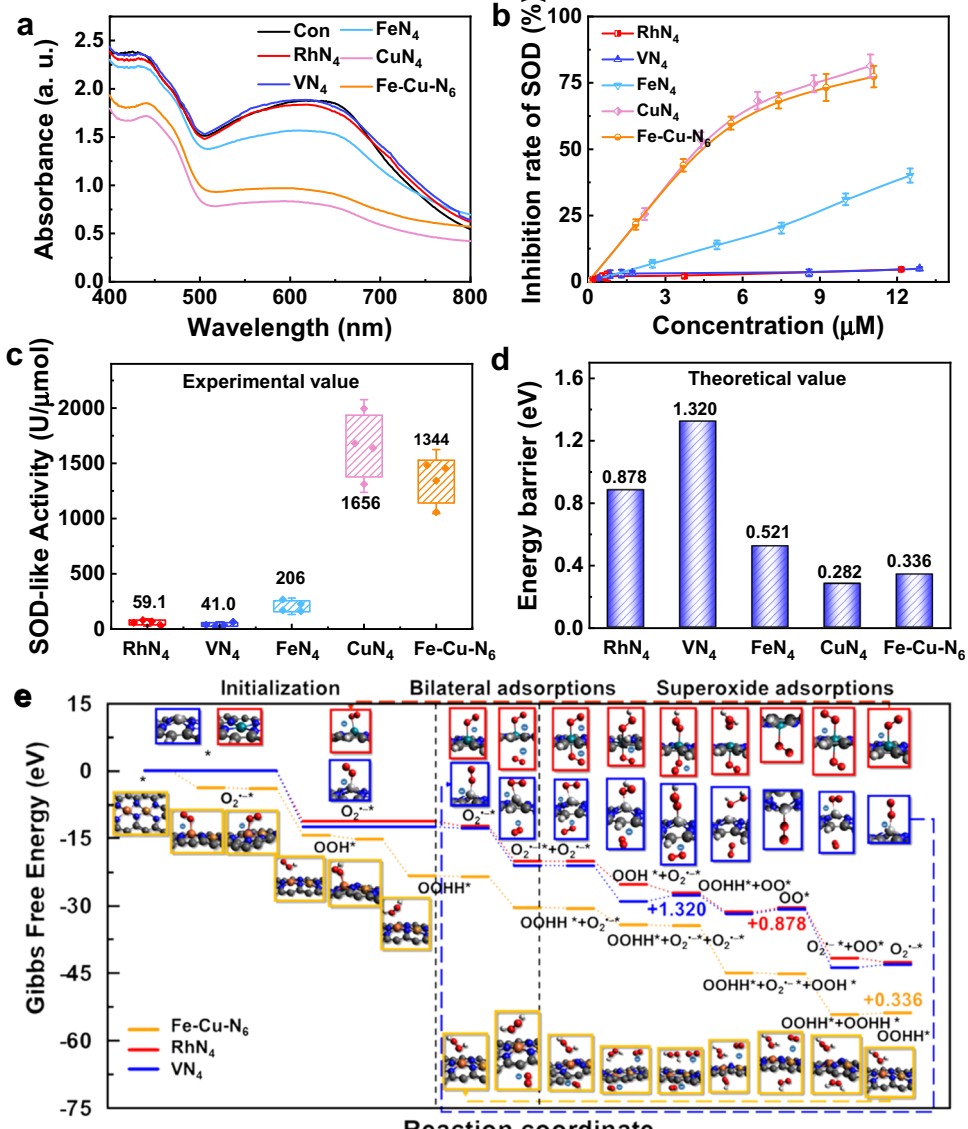

**Fig. 5 | The SOD-like activity of MN_x.** **a** SOD-like activities with and without MN_x. 'a. u.' represents arbitrary units. **b** Concentration-dependent inhibition rates of MN_x calculated by the NBT colorimetric reaction for SOD-like activities. **c** Quantification of specific SOD-like activities of MN_x ($n = 4$ independent experiments; boxes represent the median and IQR and the upper and lower whiskers extending to the values that are within $1.5 \times$ IQR). **d** The energy barriers of MN_x in the SOD-mimic reactions simulated by DFT. **e** The SOD processes with MN_x. White, dark gray, blue, red, cyan, light gray, orange, and brown balls represent H, C, N, O, Rh, V, Fe, and Cu atoms. All data are presented as mean ± SD.

ultrahigh catalytic activity of MN_x is primarily attributed to the unique "two-sided oxygen-linked" catalytic reaction path to increase the utilization of catalysts compared to the previously reported one-side reaction route[45,46]. In the catalytic procedure of VN_4, the "initialized" state enrolls an extra hydrogen atom for attachment compared to RhN_4 and Fe-Cu-N_6. The finalization step, which restores catalysts to their original states, is only observed in specific reaction pathways, including the Fe-Cu-N_6 and FeN_4 processes, the finalization process, and the reinstated initialization step of subsequent catalysis, but not in the catalytic processes of the RhN_4 and VN_4 loop between bilateral adsorption and hydrogen adsorption steps (Fig. 3e, f). In addition, the simulated POD pathways for the five possible structures of Fe-Cu-N_x showed that Fe-Cu-N_6 structure possesses the lowest energy barrier (Supplementary Fig. 22 and Supplementary Table 6), futher demonstrating Fe-Cu-N_6 is the most possible structure. Therefore, Rh/VN_4 preferentially forms the Rh/V-O-N_4 structure with an active center for the circular catalytic processes to significantly decrease reaction energy barriers, shorten the reaction path, and accelerate the catalytic kinetics, thus creating a unique

bilateral reaction path that gives rise to a catalytic performance higher than the reported FeN_4 and CuN_4.

It is worth noting that, unlike most previous studies[27,45,46], the POD-mimicking reaction pathway simulated in this work is not accompanied by the generation of hydroxyl radicals (·OH). Instead, the ·OH-producing catalytic reaction pathways were still simulated. As shown in Supplementary Fig. 23 and Supplementary Table 7, there are no significant differences in the energy barriers of FeN_4 between the two paths at 0.539 eV and 0.405 eV, suggesting the existence of reaction pathways both with and without generated ·OH. However, all the other catalysts except FeN_4 display significant differences in energy barriers between the two reaction paths, especially RhN_4 and VN_4, indicating the preference for the pathway without ·OH generation. We utilized electron spin resonance (ESR) spectroscopy to verify further whether MN_x can scavenge ·OH in the inflammatory physiological environment (pH = 6.0–6.5). It turned out that MN_x effectively scavenged ·OH (Supplementary Fig. 24). Therefore, exceedingly different from the widely reported FeN_4, the catalysis of RhN_4 and VN_4 follows

unique paths and mechanisms, forming the Rh/V-O-N$_4$ active center by preference and then activating the circulatory catalytic reaction cascade.

## CAT-like activities of MN$_x$

Meanwhile, the intrinsic CAT-like activities of MN$_x$ were evaluated by monitoring the decomposition of H$_2$O$_2$ at the concentration of 4.8 μM. Amongst all catalysts, RhN$_4$ showed the optimal catalytic performance with a clearance rate of H$_2$O$_2$ up to 100 % within 40 min (Fig. 4a and Supplementary Fig. 25). Furthermore, the CAT-like activities of RhN$_4$, Fe-Cu-N$_6$, and VN$_4$ were quantitatively determined to be 5177, 1510, and 621 U/μmol. In particular, the activity of RhN$_4$ was 5 and 15 times higher than the widely reported FeN$_4$ (1044 U/μmol) and CuN$_4$ (313 U/μmol) (Fig. 4b). Likewise, the stability of the CAT-like activity for RhN$_4$ was confirmed for over a month at room temperature, whereas the natural CAT activity was completely inactive after 14 days (Fig. 4c). Kinetic analysis revealed that RhN$_4$ had a high affinity to the H$_2$O$_2$ substrate with a $K_m$ of 1.33 mM, about 20 times more superior than the natural CAT ($K_m$ ~28.8 mM) (Supplementary Figs. 26, 27 and Supplementary Table 8). Therefore, MN$_x$ is potentially useful as an effective CAT mimetic to defend against oxidative stress in organisms during disease development.

Compared to the POD-like processes, simulated CAT-like processes are generally less complex (Fig. 4g and Supplementary Fig. 28). Without extra hydrogens, the initialization steps, including the "initialized" states and bilateral adsorption steps, are converged into the catalytic pathways of all catalysts except CuN$_4$. The "initialized" Fe, V, and Rh in the MN$_x$ limit the loop within the bilateral adsorption step. In Fe-Cu-N$_6$ with two metallic centers, the bilateral adsorption step is generalized into special adsorption on different metallic atoms. The first and second states with a single oxygen atom attached are directed to Fe (Fig. 4g row 2 col 3) and Cu (Fig. 4g row 2 col 7), respectively. FeN$_4$ and Fe-Cu-N$_6$ follow the "finalization" steps in which the catalysts are restored to their initial states. Besides, DFT simulations predict activation energies of RhN$_4$, Fe-Cu-N$_6$, VN$_4$, FeN$_4$, and CuN$_4$ to be 0.213, 0.291, 1.10, 0.539 and 1.507 eV, respectively (Fig. 4d). The RDSs associated with the energy barriers of the activation energies disclose the reaction mechanisms in-depth: RhN$_4$ breaks down the second H$_2$O$_2$ molecule with difficulty; VN$_4$ and FeN$_4$ have relatively low attractions towards the second H$_2$O$_2$ molecule; Fe-Cu-N$_6$, on the other hand, has trouble in releasing the first water molecule; CuN$_4$ can hardly extract another oxygen atom from an H$_2$O$_2$ molecule on the other side. The intermediate states of Rh/VN$_4$ and Fe-Cu-N$_6$ during catalytic cycles are consistent with those observed in the POD-like reaction pathways (Fig. 4e, f).

## SOD-like activities of MN$_x$

We then studied the SOD-like activities of MN$_x$ under the same conditions, using the classical nitro-blue tetrazolium (NBT) chromogenic approach. CuN$_4$ and Fe-Cu-N$_6$ inhibited formazan production by scavenging O$_2^{\cdot-}$ (Fig. 5a). The inhibition rate of SOD was verified via the SOD-mimic reaction assay of MN$_x$ at various concentrations (Fig. 5b). The SOD-like activities of CuN$_4$ and Fe-Cu-N$_6$ were quantitatively calculated to be 1656 and 1344 U/μmol (Fig. 5c), which were attributed to the low energy barriers predicted by DFT simulations (Fig. 5d). Therefore, the high SOD-like activity of Fe-Cu-N$_6$ can be ascribed to its Cu-N$_4$ coordination structure, similar to that of CuN$_4$. Overall, given that Fe-Cu-N$_6$ possesses both POD-like and CAT-like activities of FeN$_4$, along with the SOD-like activity of CuN$_4$, the MN$_x$ can practically achieve multienzyme-mimetic activities by integrating different metal-based active sites.

The reaction pathways with the energy diagram of the SOD-like processes are shown in Fig. 5e and Supplementary Fig. 29. The valences of the transition/intermediate states vary with the charge transfers in each reach pathway. The valence changes immediately with the

superoxide ion captured by the catalyst during the initialization steps. Instead of having a single oxygen atom attached, the "initialized" state has an attached superoxide ion as in the cases of RhN$_4$ and VN$_4$. In contrast, Fe-Cu-N$_6$, FeN$_4$, and CuN$_4$ finish the initialization steps with two more hydrogen atoms adsorbed (an H$_2$O$_2$ molecule is formed as a result) to the central metal atom (an H$_2$O$_2$ molecule is attached to the Fe atom of Fe-Cu-N$_6$). After the initialization stage, the "bilateral adsorptions" are still the primary mechanism in all catalytic processes: the "initialized" state with adsorbed O$_2^{\cdot-}$ or H$_2$O$_2$ on one side can involve further catalytic steps by attracting a second superoxide ion from the other side. After a similar bilateral adsorption process, the oxygen molecule is released, and the superoxide ion restocks, with an H$_2$O$_2$ molecule released among different catalytic processes of MN$_x$. For simplicity, we denote the five sub-steps as follows: (A) H$_2$O$_2$ formation, (B) bilateral adsorption, (C) the release of an oxygen molecule, (D) restocking of superoxide ion, and (E) the release of an H$_2$O$_2$ molecule. The reaction pathways of Rh/VN$_4$ follow the B → A → E → D → C order. The higher catalytic efficiency is likely to be ascribed to the complex formed by Rh/VN$_4$ and superoxide ions, where one electron is lost and yields a better adsorption capacity. On the other hand, the reaction pathway of Fe-Cu-N$_6$ follows the A → B → D → C → E order, which can be exclusively found in the two-center system. The unique mechanism of the Fe-Cu-N$_6$ system is distinctly different from the reaction pathways of FeN$_4$ and CuN$_4$, which follow the B→A→C→E order (Supplementary Fig. 29). The finalization step is only present in FeN$_4$ and CuN$_4$, while all the other three catalysts are looped within steps of bilateral adsorption and superoxide adsorption instead. The predicted activation energies of CuN$_4$, Fe-Cu-N$_6$, FeN$_4$, RhN$_4$, and VN$_4$ are calculated to be 0.282, 0.336, 0.521, 0.878, and 1.32 eV, respectively. Notably, DFT suggested that RDSs for CuN$_4$, Fe-Cu-N$_6$, FeN$_4$, RhN$_4$, and VN$_4$ systems are confined in the H$_2$O$_2$-related sub-steps E, E, E, A, and E, respectively.

Besides, we conducted the classic glutathione reductase (GR) coupling assay and found that only VN$_4$ had substrate concentration-dependence for the GPx-like activity, which was quantified to be 47.9 U/μmol (Supplementary Fig. 30). The reaction affinity of VN$_4$ ($K_m$ ~ 1.31 mM) to glutathione (GSH) substrate was nearly 7-fold better than that of the natural GPx ($K_m$ ~ 9.23 mM) (Supplementary Fig. 31 and Supplementary Table 9). Furthermore, the simulated GPx-mimetic reaction acquired an energy barrier of 3.28 eV during the GPx catalysis (Supplementary Fig. 32). Moreover, to exclude the interference from Zn atoms on the catalytic performance of MN$_x$, the multiple enzyme-like activities of ZIF-N/C only anchored by Zn atoms were also tested (Supplementary Fig. 33). The ZIF-N/C showed a negligible catalytic activity, validating that the ultrahigh catalytic properties predominantly originate from the M-N$_x$ active centers. In addition, the Donor Fukui Function (defined in Eq. 37) manifests that the electrophilic reactivities of the four central metals are significantly different, implying different charge-donating abilities of active sites on RhN$_4$, VN$_4$, and Fe-Cu-N$_6$ (Supplementary Fig. 34)

## MN$_x$-based sutures for scalp healing in traumatic brain injuries

Inspired by the enzyme-mimicking activities of MN$_x$ theoretically and experimentally demonstrated above, we developed MN$_x$ sutures by integrating MN$_x$ into polyglycolic acid (PGA) sutures and applied them to the scalps of living mice with traumatic brain injuries (TBI) to study their potential healing effects. The field-emission scanning electron microscopic (FE-SEM) images and ICP-MS showed that MN$_x$ was successfully incorporated into the suture structures (Supplementary Figs. 35, 36 and Supplementary Table 10). Meanwhile, the MN$_x$ sutures retained the original high enzyme-like activities, with consistent catalytic performances (Supplementary Fig. 37). Furthermore, as demonstrated by XRD and Raman spectroscopy, the characteristic peaks of graphite carbon were well preserved after modification of sutures with MN$_x$, despite the interference from the inherent PGA background

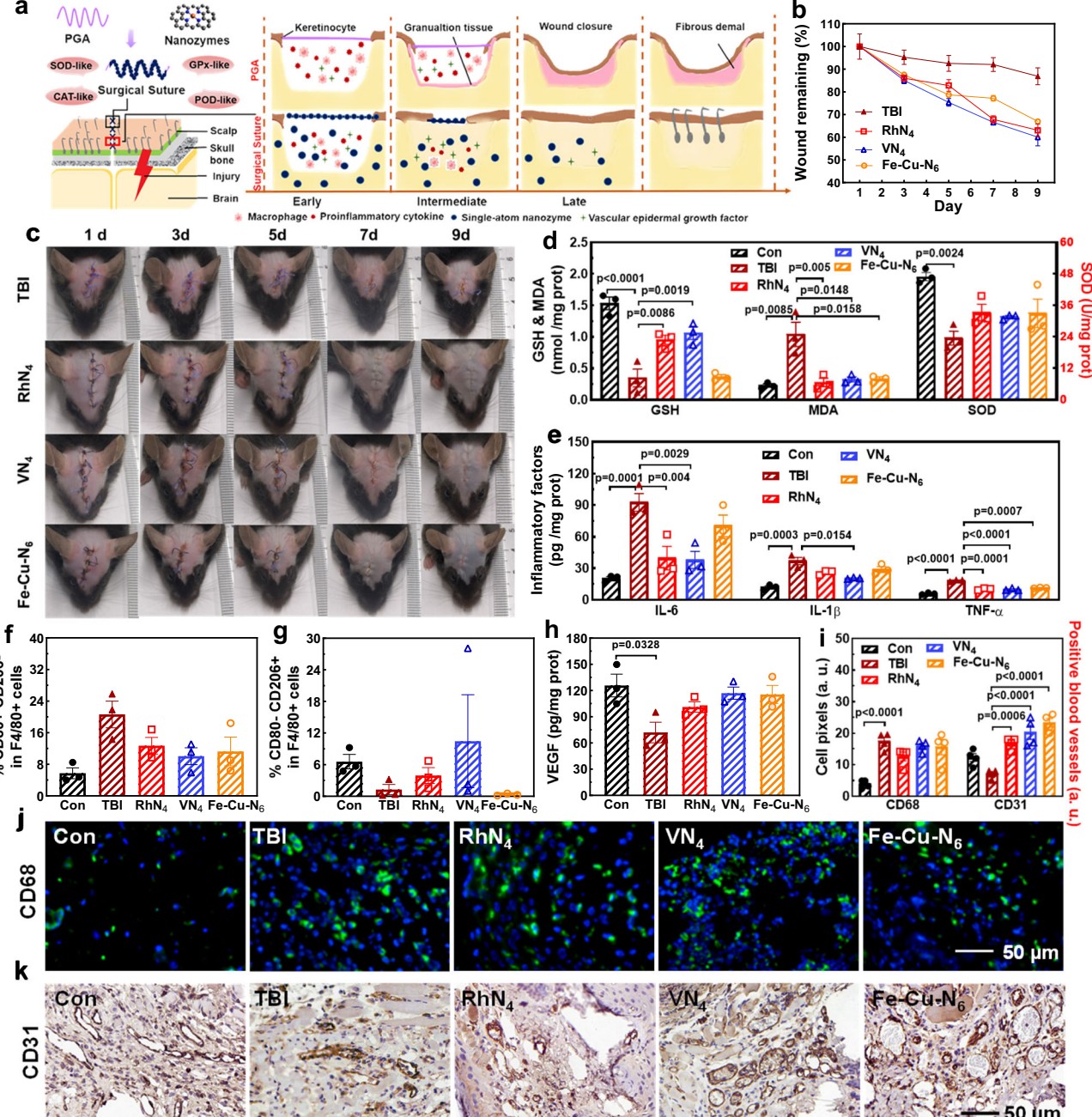

**Fig. 6 | The scalp healing process with MN$_x$ sutures. a** Schematic illustration of the scalp closure process modulated by MN$_x$ sutures. **b, c** Representative residual wound (**b**) and photographs of scalp healing (**c**) over time with and without MN$_x$ sutures ($n$ = 3 images/3 mice). **d** Indicators for oxidative stress of the scalp on day 3 post injury ($n$ = 3 biologically independent samples). **e** ELISA analysis of IL-6, IL-1β and TNF-α levels in scalp with or without MN$_x$ sutures on day 3 post-injury ($n$ = 3 biologically independent samples). **f, g** The flow cytometry analysis of **f** M1 and **g** M2 macrophages in the scalp after treatments ($n$ = 3 biologically independent samples). **h** ELISA analysis of VEGF levels in scalp with or without MN$_x$ sutures on day 3 post-injury ($n$ = 3 biologically independent samples). **i, j** Immunofluorescence staining of **j** CD68 and **i** the quantitative analysis in the injured scalp on day 4 post wound injury ($n$ = 4 images/4 mice). **k** Immunohistochemistry of CD31 and **i** the quantitative analysis in scalp on day 4 post wound injury ($n$ = 4 images/4 mice). Data are presented as mean ± standard error of the mean and compared with the TBI by one-way ANOVA with the one-sided Tukey's multiple comparisons test (the $p$-values are shown). Differences with $p$-values < 0.05 are considered significant.

signal (Supplementary Figs. 38, 39). Additionally, signals of aggregated metal nanoparticles were not observed, suggesting the single-atomic distribution in the suture framework available for efficient catalysis.

To explore the medical practicability of MN$_x$, we employed mouse models of TBI with skin wounds to investigate the in vivo wound-healing effects. Wound healing is a complex physiological process divided into four stages: hemostasis, inflammation, proliferation, and tissue modeling, which have complicated and dynamic interactions

associated with time. Figure 6a depicts the schematic illustration for the wound closure process modulated with MN$_x$ stitches. We hypothesized that the slow healing of scalp injuries using pristine PGA sutures could be accelerated by using MN$_x$ sutures by inhibiting the secretion of inflammatory cytokines and harmful molecules. As shown in Fig. 6b, c, and Supplementary Fig. 40, MN$_x$-modified surgical sutures successfully seamed the wound with time. In addition, the wound size and area were significantly diminished and healed to healthy levels

with $MN_x$ sutures, whereas the untreated TBI mice only showed a partial recovery of about 80%. As shown in Fig. 6d, Supplementary Figs. 41, 42, indicators of malondialdehyde (MDA), SOD, GSH, and $H_2O_2$ in the scalp tissues were relatively severe in the TBI group. By contrast, the decrease in SOD and GSH levels from scalp injuries could be rescued by $MN_x$ sutures with prominent recoveries on day 3 after treatment. Comparatively, $VN_4$ induced a better recovery in GSH than $RhN_4$ and $Fe\text{-}Cu\text{-}N_6$, which correlated well with their in vitro GPx-like activities. Lipid peroxides and $H_2O_2$ can continuously accumulate in the scalp following injuries as the by-products of oxidative stress, resulting in severe damage. All $RhN_4$, $VN_4$, and $Fe\text{-}Cu\text{-}N_6$ considerably inhibited the production of these harmful molecules and maintained their stable POD-like and CAT-like activities in the long term.

Excessive oxidative indicators, such as proinflammatory cytokines and macrophages, can prolong the inflammation duration and jeopardize the transition into the proliferative phase[57,58]. From enzyme-linked immunosorbent assays (ELISA), proinflammatory cytokines such as interleukin (IL)-6, IL-1β, and tumor necrosis factor-α (TNF-α) were significantly upregulated following wound injuries, indicative of potent local inflammation (Fig. 6e and Supplementary Fig. 43a–c). Fortunately, $RhN_4$ and $VN_4$ could sharply downregulate IL-6, IL-1β, and TNF-α levels, with a notable anti-inflammation effect. In comparison, $Fe\text{-}Cu\text{-}N_6$ only showed negligible regulation of IL-6 and IL-1β due to its weaker catalytic activity. Although PGA sutures could gradually suppress the inflammatory cytokines through autoimmunity, $MN_x$ treatment essentially restored the cytokines to normal levels, suggesting its role in suppressing neuroinflammation. As shown in Fig. 6f, g, Supplementary Figs. 44–47, the population of M1 macrophages decreased and was polarized into the M2 phenotype, further confirming the anti-inflammatory effects of $MN_x$ sutures. In addition, changes in other immune cells like natural killer cells, B cells, and T cells after various treatments in whole blood were also analyzed by flow cytometry. As shown in Supplementary Fig. 48, mice could induce systemic immune responses after brain injuries and gradually recovered to normal with $MN_x$-based sutures. Besides, the tissue macrophage marker CD68 was upregulated after scalp injuries but recovered by $MN_x$ sutures, while the vascular epidermal growth factor (VEGF) was also ameliorated by $MN_x$ sutures (Fig. 6h–j, Supplementary Figs. 43d, 49). VEGF is known to benefit vascular endothelial cell migration and angiogenesis. $VN_4$ and $Fe\text{-}Cu\text{-}N_6$ exhibited marginally improved effects on VEGF than $RhN_4$, with potentially higher healing efficacy. Moreover, the vascular endothelial-specific marker CD31 indicated neovascularization in the injured sites. Compared with other groups, more neovascularization was observed for $VN_4$ and $Fe\text{-}Cu\text{-}N_6$ than $RhN_4$ by confocal microscopy, which directly correlated with the VEGF expressions (Fig. 6i, k, and Supplementary Fig. 50). In the early stages of wound healing, the transition from the inflammatory stage to the proliferative stage was accelerated by $MN_x$ sutures, with granulation tissues observed in the treatment groups, indicating $MN_x$ can competently mediate a beneficial wound healing process (Supplementary Fig. 51)[59,60]. Accordingly, we further investigated the effects of $MN_x$ sutures on oxidative stress and inflammation in TBI. All indicators exacerbated towards intense local inflammation and oxidative stress following TBI. However, $MN_x$ could sharply reduce harmful molecules and proinflammatory cytokines and oppositely rescue beneficial molecules in the brain (Supplementary Figs. 52, 53). Immunofluorescence (IF) staining of the cerebral cortex showed that $MN_x$ could remarkably decrease TBI-elevated expressions of IL-1β and TNF-α, inhibit microglial activation, rescue neurons (Supplementary Figs. 54, 55), and restore the body weight loss (Supplementary Fig. 56). In addition, no metal elements were detected in the blood of $MN_x$-treated mice on day 6 after injuries, suggesting that the $MN_x$-based sutures are biologically safe (Supplementary Fig. 57).

## Discussion

Rational design and practical exploration of nanozymes fulfill the unmet need for sustained and efficient enzymatic catalysis, a category that naturally derived enzymes have not yet covered. Currently, the inferior catalytic activities of nanozymes are primarily assigned to the low efficiency of atom utilization and scarcity of locally active sites. In the current work, $MN_x$ nanozymes enable precise modulation of the central coordination atom to improve overall activities catalytically outperforming natural enzymes. Besides, we identified a unique mechanism of "bilateral adsorption" from which the high enzyme-mimicking catalytic activity originated. During this process, the $X^*\text{-}M\text{-}X^*$ ($X = O, OH, O_2^{\bullet-}$) bilateral adsorption structure first formed and then reduced the activation energies, facilitating fine-tuning the coordination microenvironment of the central metal atom, which is necessary to boost the entire catalytic cycle. Hence, we deduced the $X^*\text{-}M\text{-}X^*$ ($X = O, OH, O_2^{\bullet-}$) intermediate state as the cyclic block of catalysis with a proposed "bilateral adsorption" mechanism.

As is known, the catalytic activities of naturally-occurring enzymes are unstable and would decay within 2 days, while $MN_x$ nanozymes behave in a comparatively stable manner without any evident decay after 1 month (Figs. 3c, 4c, and Supplementary Fig. 58). Another critical challenge is that natural enzymes can hardly be repeatedly used. They are efficient and versatile as biocatalysts in nature but suffer from environmental disturbance, poor catalytic stability, and low sustainability, making them limited for widespread use under current circumstances. By comparison, the catalytic activities of $MN_x$ nanozymes were barely attenuated during repetitive catalytic processes (Supplementary Fig. 59). All these advantageous features justified $MN_x$ as a promising enzyme substitute in the case of continuous biocatalysis. As shown in the schematic wound closure process (Fig. 6a), scalp injuries from brain trauma are physiologically complex, accompanied by complicated and dynamic interactions strongly associated with time, and can continuously trigger a cascade of molecular and biochemical events in vivo, resulting in severe inflammation and oxidative stress[59,61]. Typically, the scalp injuries with PGA sutures only recovered slowly from the inflammatory phase to the tissue modeling. Nevertheless, the as-developed $MN_x$ nanozymes with definitive molecular structures circumvent all such obstacles and accelerate the healing process with recyclable and catalytic features, remedying the limitations of natural enzymes.

In summary, we developed single-atom nanozymes of $RhN_4$, $VN_4$, and $Fe\text{-}Cu\text{-}N_6$ with $M\text{-}N_4$ active centers and achieved efficient multienzyme-mimetic catalysis with good selectivity by maximizing atom utilization and tuning substitution atoms. $RhN_4$ and $VN_4$ preferentially exhibited high POD-like activities with $K_m$ of 55.7 and 65.6 μM to the TMB substrate, ~5-fold lower than the natural enzyme counterpart HRP ($K_m \sim 276.2$ μM). In contrast, $RhN_4$ ($K_m \sim 1.33$ mM) and $Fe\text{-}Cu\text{-}N_6$ ($K_m \sim 1.99$ mM) showed 20 and 14 times higher affinities to $H_2O_2$ substrate than the natural CAT ($K_m \sim 28.8$ mM) during CAT-mimetic reactions. Further, $Fe\text{-}Cu\text{-}N_6$ exhibited a high SOD-like activity of 1344 U/μmol, while $VN_4$ possessed the unique GPx-like activity with an affinity to GSH ($K_m \sim 1.31$ mM), exceeding that of natural GPx ($K_m \sim 9.23$ mM). Meanwhile, DFT calculations pinpointed a unique bilateral reaction mechanism that reduced the reaction energy barriers and thus enhanced catalytic activities. Further, we medically applied $MN_x$ sutures that accelerated scalp healing from brain trauma via promoting VEGF and inhibiting oxidative stress and neuroinflammation.

## Methods

### Chemicals

Rhodium (III) 2,4-pentanedionate ($Rh(acac)_3$, 98%) was purchased from Shanghai Xinbo Chemical Technology Co., Ltd. (China). Zinc nitrate hexahydrate ($Zn(NO_3)_2 \cdot 6H_2O$, 99%), hydrogen peroxide (30 wt % in $H_2O$), Dimethyl sulfoxide (DMSO, >99.8%), and isopropanol (99.8%) were purchased from Shanghai Aladdin Biochemical Co., Ltd.

(China). Vanadium (III)-2,4-pentanedionate (V(acac)$_3$, 97%), 2-methylimidazole (99%), and 3,3′,5,5′-tetramethylbenzidine (TMB, 99%) were purchased from Meryer (Shanghai) Chemical Technology Co., Ltd. (China). Iron (III) chloride hexahydrate (FeCl$_3$·6H$_2$O, 99%), Cupric chloride (CuCl$_2$, 98%), N, N-dimethylformamide (DMF, 99.5%), and HRP (>300 U/mg), were obtained by Shanghai Macklin Biochemical Co., Ltd. (China). GPx (740 U/mg), CAT (4323 U/mg) were purchased from Sigma-Aldrich Company Ltd. NADPH tetracyclohexanamina (90%), GR (1164.8 U/mL) were purchased from Shanghai Yuanye Bio-Technology Co., Ltd. (China). GSH (97%) was purchased from Tianjin Xiensaopude Technology Co., Ltd. (China). SOD (5200 U/mg) was purchased from Beijing Solarbio Science & Technology Co., Ltd. (China). Deionized water (>99.5%), methanol (>99.5%), and ethanol absolute (>99.7%) were purified to analytical grade by Tianjin Yuanli Chemical Co., Ltd. (China). Except for special instructions, all experiments were carried out at room temperature.

## Preparation of MN$_x$

The VN$_4$ single-atom enzyme mimetics were synthesized by a one-pot method. Firstly, 0.8733 g of 2-methylimidazole was added to 20 mL of methanol to form a transparent solution. Next, the 77.3 mg of V(acac)$_3$ and 0.8032 g of Zn(NO$_3$)$_2$·6H$_2$O were dissolved in a beaker containing 10 mL of methanol to form the dark green solution. The two solutions were mixed and stirred vigorously. Then, the mixture was poured into a 50 mL high-pressure reactor lined with polytetrafluoroethylene, heated instantly to 120 °C, and kept for 4 h. The gray precipitate was collected and then naturally cooled to room temperature. It was then purified several times with DMF and anhydrous methanol, respectively, and V-doped ZIF-8 (V/ZIF-8) was obtained. Next, the dried V/ZIF-8 precursor powder was heated to 900 °C at a rate of 5 °C/min in a tube furnace under nitrogen gas flow and kept for 3 h. The resulting black powder was the VN$_4$. Similarly, RhN$_4$ follows similar procedures, and the difference is the use of 88 mg of Rh(acac)$_3$ instead of 77.3 mg of V(acac)$_3$.

The FeN$_4$ was prepared by a two-step method. First, to synthesize the carbon-based supporting substrate, 5.909 g of 2-methylimidazole was dissolved in a beaker A containing 450 mL of methanol. Next, 5.085 g of Zn(NO$_3$)$_2$·6H$_2$O was uniformly dissolved in another beaker with 500 mL of methanol and poured into beaker A. The mixture was rapidly stirred until it turned cloudy white and placed in a drying oven at 60 °C for 24 h. The white precipitate was collected by centrifugation and washed five times with absolute ethanol to obtain transparent ZIF-8 nanocrystals. After drying overnight in a vacuum, the nanocrystals were pyrolyzed at 900 °C for 2 h in an annealing furnace under nitrogen gas flow. The final product was the N-doped carbon framework with a rich pore structure, marked as ZIF-C/N.

The preparation of FeN$_4$ and CuN$_4$ was obtained by activating ZIF-C/N that was embedded with different metal ions at high temperatures. Typically, 40 mg ZIF-C/N powder was dispersed in isopropanol (4 mL) containing 2 mg FeCl$_3$. The mixture was then sonicated and stirred vigorously for 2 h, resulting in Fe$^{3+}$ adsorbed in the ZIF-C/N. Then, the ZIF-C/N with Fe$^{3+}$ was activated in a tube-type furnace at 500 °C under a nitrogen atmosphere for 1 h, and the final product was the single-atom FeN$_4$. CuN$_4$ followed similar procedures, and the difference was the use of 1.5 mg CuCl$_2$ instead of 2 mg FeCl$_3$.

Similarly, the synthetic procedure of Fe-Cu-N$_6$ was N$_2$-annealing (500 °C) activation of ZIF-C/N co-embedded with Fe$^{3+}$ and Cu$^{2+}$ ions, in which the feeding of metal ions was 1 mg FeCl$_3$ and 0.75 mg CuCl$_2$.

## Preparation of catalyzed MN$_x$ sutures

The PGA sutures were purchased from Shandong Haidike Medical Products Co., Ltd. Each PGA suture was individually fixed on a homemade iron frame and then immersed in an isopropanol solution of MN$_x$ for 3–5 days so that MN$_x$ was uniformly loaded on the sutures.

Meanwhile, the MN$_x$ solution was sonicated once every 4 h during the soaking procedure until the MN$_x$ was evenly dispersed. Last, the MN$_x$ sutures were dried overnight in a vacuum oven for subsequent characterization and activity testing.

## Characterization

The morphology and size of MN$_x$ were observed by bright-field TEM and corresponding EDS element mapping images on two field-emission transmission electron microscopes (JEM-F200 and JEM-2100F, JEOL, Japan) with an accelerating voltage of 200 kV. Next, the images were taken on a cooled emission spherical aberration-corrected atomic resolution microscope (JEM-ARM200F, JEOL, Japan) running at a resolution of 0.08 nm and a working voltage of 200 kV. Finally, the samples were scattered on the micro-grid carbon grid.

The XAFS spectra were obtained at the 1W1B station (2.5 GeV, maximum current of 250 mA) of the Beijing Synchrotron Radiation Facility. The ATHENA and ARTEMIS software were used to analyze and fit the XAFS results. All samples' ultraviolet-visible (UV-vis) absorption spectra were tested on the UV-VIS NIR instrument (Shimadzu, Japan). The micro-quartz cuvette with a light path of 1 cm was utilized to contain the solution. The crystal structure of the precursor, the final MN$_x$ samples, and MN$_x$ sutures were researched by X-ray diffraction spectroscopy (XRD) scanning on an X-ray powder diffractometer (Smartlab, Rigaku, Japan) radiated by Cu Kα. The Raman spectra of carbon-based MN$_x$ and PGA sutures loaded with MN$_x$ were measured by an ultra-fast Raman imaging spectrometer (XploRA PLUS, Horiba-JY, France) with the scanning wavenumber range of 500 ~ 3800 cm$^{-1}$. The valence states of different elements were detected by an X-ray photoelectron spectrometer (ESCALAB Xi$^+$, Thermo Fisher Scientific, UK) excited by Al Kα, in which powder samples were pressed into discs. The original data were peak-fitted in the XPSPEAK41 software. SEM and elemental mapping images were collected using a field-emission scanning electron microscope for the MN$_x$-based PGA sutures (S-4800, Hitachi, Japan). The ICP-MS measurement (7900 ICP-MS, Agilent, UK) was employed to characterize the content of metal elements in MN$_x$ and MN$_x$-sutures.

## The POD-like activity test

The quantitative detection of the POD-like activities of the MN$_x$ was tested by colorimetric method using a TMB substrate color development kit (TELISA, SenBeiJia). During the experiment, the working solution was prepared according to the instructions. First, MN$_x$ samples (20 μL) and the working solution (180 μL) were added to the 96-well plate. Then, the absorbance changes at 652 nm were monitored over time through a microplate spectrophotometer (CMax Plus, Molecular Devices). Meanwhile, the different concentrations of HRP were selected as the standard. At last, the activity value of natural HRP and the corresponding A$_{652}$ were plotted and fitted in the Origin software to obtain a standard curve. Then, based on the standard curve, the POD-like activity of the MN$_x$ was quantitatively calculated according to A$_{652}$.

The kinetic test of the interaction between the MN$_x$ and TMB substrate was carried out using a UV-vis spectrophotometer under the kinetic mode to record the increase at A$_{652}$ of the reaction solution over time. In a typical assay, the different concentrations of TMB (0–800 μM, dissolved in DMSO) as varying substrate, H$_2$O$_2$ solution (50 mM), and MN$_x$ (or HRP) were added to sodium acetate buffer (PH = 4.5) to maintain the volume of the mixture at 200 μL. Then, the increase at A$_{652}$ was immediately recorded to measure the corresponding initial rate. The maximum reaction velocity ($V_m$) and $K_m$ were determined by plotting the initial rate and the substrate concentration using Microsoft Excel. Curves were fitted to the Michaelis−Menten curve on the Origin software. The $k_{cat}$ represents the catalytic

efficiency of $MN_x$ per molar concentration, and its value was obtained by dividing $V_m$ by the concentration of $MN_x$ (E) during the test. The concentration of $MN_x$ was expressed as the total concentration of active metal sites measured by ICP-MS.

In addition, kinetic analysis between $MN_x$ and $H_2O_2$ substrate was studied by measuring the changes at $A_{652}$ in the reaction mixture with different concentrations of $H_2O_2$ (0–50 mM) at a fixed TMB (800 μM) concentration.

## The CAT-like activity assay

The CAT-like activity of $MN_x$ was assessed by measuring the decomposition rate of $H_2O_2$. In short, $MN_x$ of the same concentration (4.8 μM) was added to 40 μM $H_2O_2$ aqueous solution (300 μL), then the absorbance change at 240 nm over time was detected on a spectrophotometer. Moreover, a mixture of 100 μM $H_2O_2$ aqueous solution (380 μL) and $MN_x$ was incubated in a centrifuge tube for about 3 min. The picture containing abundant bubbles was captured, indicating that $MN_x$ catalyzed $H_2O_2$ to produce $O_2$.

The quantitative detection of the CAT-like activity of the $MN_x$ was obtained by using the Catalase Assay kit (S0051, Biyuntian). One unit of the enzyme activity was defined as 1 μmol $H_2O_2$ decomposed within 1 min at 25 °C under neutral conditions (pH = 7).

Further, the kinetic analysis of $MN_x$ was performed using a Dissolved Oxygen Meter (HACH HQ40d, USA) with an LDO101 probe to measure the rate of oxygen generation catalyzed by $MN_x$ at different concentrations of $H_2O_2$ (0–5 mM). A Michaelis–Menten curve was established based on the $O_2$ production rate and the $H_2O_2$ concentration and fitted to derive $K_m$ and $V_m$. For the kinetic test of the natural CAT, the decrease in absorbance at 240 nm was monitored at different concentrations of $H_2O_2$ (0–40 mM) in the presence of CAT using a UV-vis spectrophotometer in the kinetic mode. The molar extinction coefficient $\varepsilon_{240 nm}$ of $H_2O_2$ was calculated to be 39.4 $M^{-1}$ $cm^{-1}$.

## The SOD-like activity assay

The SOD-like activity of $MN_x$ was determined using the Total Superoxide Dismutase Assay Kit with NBT (R22261, Yuanye). In a typical assay, the SOD detection buffer (100 μL), NBT color developing solution (30 μL), an enzyme solution (30 μL), and different concentrations of $MN_x$ (20 μL) were added to a 96-well plate. Then, the reaction starter solution (20 μL) was added. The reaction mixture was irradiated under a fluorescent lamp for an appropriate amount of time, and the absorbance at 560 nm was measured immediately with a microplate spectrophotometer (CMax Plus, Molecular Devices). Simultaneously, a 20 μL SOD detection buffer was used instead of $MN_x$ as the blank control group ($A_{con1}$), and the other control group used an SOD detection buffer (20 μL) instead of $MN_x$ in the dark during the experiment as a light-proof control group ($A_{con2}$). The $A_{560}$ was converted into the inhibition rate of the sample through the formula: inhibition rate (%) = ($A_{con1}$ − $A_{MNx4}$) / ($A_{con1}$ − $A_{con2}$) × 100. The SOD-like activity of $MN_x$ was further calculated according to the formula: enzyme activity (units) = inhibition percentage / (1 − inhibition percentage), in which 50% of inhibition on NBT photochemical reduction was taken as an enzyme activity unit.

## The GPx -like assay

According to the previously reported glutathione reductase coupling reaction, the GPx-like activity of the $MN_x$ was evaluated by monitoring the decrease in NADPH content after $MN_x$ treatment. In the assay, the detection solution was composed of phosphate buffer (PBS, pH = 7.4), $MN_x$ solution (3.8 μM), reduced glutathione (GSH, 2 mM), glutathione reductase (GR, 1.7 Units/mL), NADPH (200 μM), and $H_2O_2$ (0.2 mM) at the total volume of 500 μL. When $H_2O_2$ was added, the change of DADPH at 340 nm ($\varepsilon_{340 nm}$ = 6.22 $mM^{-1}$ $cm^{-1}$) was immediately measured within

5 min using a UV-vis spectrophotometer operating in the kinetic mode. Furthermore, kinetic measurements of the interactions between $MN_x$ (or natural GPx) and GSH as a substrate were achieved at various concentrations of GSH (0–5 mM), while the $H_2O_2$ concentration was fixed at 0.2 mM. Conversely, the kinetic analysis of substrate $H_2O_2$ was conducted by changing the concentration of $H_2O_2$ with the fixed GSH (2 mM). Finally, the calculation process of $K_m$ and $V_m$ is consistent with the method mentioned above.

The quantitative detection of GPx-like activities of $MN_x$ was carried out under strict conditions of 25 °C by using a total glutathione peroxidase detection kit (S0058, Beyuntian). One enzyme activity unit (1 unit) was defined as 1 μmol NADPH converted into $NADP^+$ within 1 min catalyzed by $MN_x$ in the presence of GSH, GR, and organic peroxide (Cum-OOH) at 25 °C and pH 8.0.

## ˙OH scavenging test

The ˙OH scavenging measurements were conducted on an ESR spectrometer (JES-FA200, JEOL, Japan). The nitrogen trap DMPO was used to trap ˙OH to form DMPO-OH spin adducts. During the assay, ˙OH was generated by UV illumination of a centrifuge tube containing DMPO (50 mM), $H_2O_2$ (2 mM), and acetate buffer (pH = 6.1) for about 20 min. Then, the mixture was transferred into a quartz tube for ESR measurement, and a clear 4-peak signal with an intensity of 1:2:2:1 was captured. Subsequently, $MN_x$ (100 ng/μL) was added, and the ESR test was performed again immediately.

## DFT simulation

Generally, a reaction is not only governed in the direction of energy minimization but is also significantly affected by the reaction rate. An exothermic reaction is energetically favored in nature but slow in practice. Therefore, stepping down in Gibbs free energy is more meaningful for chemical reactions. As we follow the changes in Gibbs free energies along the reaction path, the bottlenecks are always the ascending changes that catalytic materials could significantly reduce. The catalysts may involve many intermediate and transition states by adsorbing reactants or fragments. Many attempts were made to determine the reaction path and the catalytic mechanism to approach the reaction intermediates and transition states. The intermediates and transition states were simulated by DFT using the Gaussian 09 package[62] and fukui function analysis using the Multifunctional Wavefunction Analyzer[63]. 6−31 G* and LAN2TZ54 were used to reduce the computation workload as the basis sets for light and heavy atoms[64]. Since DFT approximated the electron-electron interactions as a background functional of electronic density, the functional was generally split into 2 parts: the Hartree potential and the exchange-correlation potential. We chose the well-known B3LYP[65] as the exchange-correlation potential. We did not assume close shells for different states but considered multiple possibilities of spins in this work. For all systems, the corresponding zero-point-energy (ZPE) and thermal corrections of frequency analyses were performed at the same level of theory to obtain Gibbs free energy.

Based on the experimental results, the model catalyst was constructed as a graphene flake with its central region replaced by a single metal atom surrounded by 4 N atoms[52]. The model catalyst was positively charged according to the experimental results and literature[66]. With the structure, the intermediate state could be obtained by fully relaxing the geometry of the flake with fragment/reactant attached. The transition state was the saddle point along the reaction coordinate.

Experimentally, different catalysts showed POD, CAT, and SOD-mimicking activities. Therefore, we focused on exploring POD-like, CAT-like, and SOD-like processes in the simulation.

## CAT-like process of MN$_x$

With the simulated reaction pathways, the procedure of the CAT-like reaction can be summarized as:

$$H_2O_2 + MN_x = MN_x - O + H_2O \tag{1}$$

which is the initialization of a catalytic process. The catalytic molecule is activated as $MN_x$–$O$ complex. The reaction then behaves as

$$H_2O_2 + MN_x - O = MN_x - O - O + H_2O \tag{2}$$

The $MN_x$–$O$–$O$ complex represents the "bilateral adsorptions" of the MN$_x$ with one oxygen atom on each side. Then another H$_2$O$_2$ molecule can be broken down into water and oxygen.

$$H_2O_2 + MN_x - O - O = MN_x - O + H_2O + O_2 \tag{3}$$

RhN$_4$ and VN$_4$ repeat Eqs. (2) and (3), but Fe-Cu-N$_6$ continues to the finalization step:

$$H_2O_2 + MN_x - O = MN_x + H_2O + O_2 \tag{4}$$

Then, Fe-Cu-N$_6$ returns to the initialization step (Eq. (1)) for the next cycle of catalytic procedures.

## The POD-like process of MN$_x$

The POD-like mechanism is more complicated than the CAT-like for more units joint the reaction, and there are more possibilities in the intermediate/transition states. The total reaction of the POD-like mechanism is:

$$2H + H_2O_2 = 2H_2O \tag{5}$$

Due to the similarities between the CAT and POD processes, the initialization steps are the same as the CAT process (see Eq. (1)). Therefore, to be precise, we provide all the reaction steps by MN$_x$. RhN$_4$ has the following POD-like procedure of reaction:

$$H_2O_2 + RhN_4 = RhN_4 - O + H_2O \tag{6}$$

$$H_2O_2 + RhN_4 - O = RhN_4 - O - O + H_2O \tag{7}$$

$$H + RhN_4 - O - O = RhN_4 - O - OH \tag{8}$$

$$H + RhN_4 - O - OH = RhN_4 - OH - OH \tag{9}$$

$$H + RhN_4 - OH - OH = RhN_4 - OH + H_2O \tag{10}$$

$$H_2O_2 + RhN_4 - OH = RhN_4 - OH - OOHH \tag{11}$$

$$H + RhN_4 - OH - OOHH = RhN_4 - O + 2H_2O \tag{12}$$

The main loop of the catalytic process is from Eqs. (7)–(12). VN$_4$ has the following POD-like procedure of reaction:

$$H_2O_2 + VN_4 = VN_4 - O + H_2O \tag{13}$$

$$H + VN_4 - O = VN_4 - OH \tag{14}$$

$$H_2O_2 + VN_4 - OH = VN_4 - O - OH + H_2O \tag{15}$$

$$H + VN_4 - O - OH = VN_4 - O + H_2O \tag{16}$$

The main loop of the catalytic process is from Eqs. (14)–(16). Fe-Cu-N$_6$ has the following POD-like procedure of reaction:

$$H_2O_2 + (Fe - Cu - N_6) = (Fe - Cu - N_6) - O + H_2O \tag{13}$$

$$H_2O_2 + (Fe - Cu - N_6) - O = (Fe - Cu - N_6) - O - O + H_2O \tag{14}$$

$$H + (Fe - Cu - N_6) - O - O = (Fe - Cu - N_6) - O - OH \tag{15}$$

$$H + (Fe - Cu - N - C) - O - OH = (Fe - Cu - N_6) - O + H_2O \tag{16}$$

$$H + (Fe - Cu - N_6) - O = (Fe - Cu - N_6) - OH \tag{17}$$

$$H + (Fe - Cu - N_6) - OH = (Fe - Cu - N_6) + H_2O \tag{18}$$

## SOD-like process of MN$_x$

The equation of reactions of SOD-like activities for Rh/VN$_4$ can be summarized in the following steps:

$$O_2^- + \text{Rh/VN}_4(III) = \text{Rh/VN}_4(II) - OO \tag{19}$$

$$O_2^- + \text{Rh/VN}_4(II) - OO = \text{Rh/VN}_4(I) - OO - OO \tag{20}$$

$$H^+ + \text{Rh/VN}_4(I) - OO - OO = \text{Rh/VN}_4(II) - OO - OOH \tag{21}$$

$$H^+ + \text{Rh/VN}_4(II) - OO - OOH = \text{Rh/VN}_4(III) - OO + H_2O_2 \tag{22}$$

$$O_2^- + \text{Rh/VN}_4(III) - OO = \text{Rh/VN}_4(II) - OO + O_2 \tag{23}$$

The Fe-Cu-N$_6$ has a SOD-like process with the initialization stage which can be summarized as the following equations:

$$O_2^- + (Fe - Cu - N_6)(I) = (Fe - Cu - N_6)(0) - OO \tag{24}$$

$$H^+ + (Fe - Cu - N_6)(0) - OO = (Fe - Cu - N_6)(I) - OOH \tag{25}$$

$$H^+ + (Fe - Cu - N_6)(I) - OOH = (Fe - Cu - N_6)(II) - OOHH \tag{26}$$

The "bilateral adsorptions" and "superoxide adsorptions" react as the following procedures:

$$O_2^- + (Fe - Cu - N_6)(II) - OOHH = (Fe - Cu - N_6)(I) - OO - OOHH \tag{27}$$

$$O_2^- + (Fe - Cu - N_6)(I) - OO - OOHH = (Fe - Cu - N_6)(0) - OO - OO - OOHH \tag{28}$$

$$H^+ + (Fe - Cu - N_6)(0) - OO - OO - OOHH = (Fe - Cu - N_6)(I) - OO - OOH - OOHH \tag{29}$$

$$H^+ + (Fe - Cu - N_6)(I) - OO - OOH - OOHH$$
$$= (Fe - Cu - N_6)(II) - OO - OOHH + H_2O_2 \tag{30}$$

$$(Fe - Cu - N_6)(II) - OO - OOHH = (Fe - Cu - N_6)(II) - OOHH + O_2 \tag{31}$$

## GPx-like process of VN$_4$

The GPx-like activity is similar to POD activities. The GSH provides a hydrogen atom to $H_2O_2$ via catalyst. The total reaction is

$$2GSH + H_2O_2 = GSSG + 2H_2O \tag{32}$$

The GPx reaction can be decomposed in the following steps with catalyst involved:

$$H_2O_2 + VN_4 = VN_4 - OOHH \tag{33}$$

$$GSH + VN_4 - OOHH = VN_4 - OH - GS \tag{34}$$

$$GSH + VN_4 - OH - GS = VN_4 - H_2O + GSSG \tag{35}$$

## Fukui function

Fukui function[67] is an important concept and a valid approach for analysis of DFT, which can be used in predicting the reactivity of electrophilic and nucleophilic at different sites[68]. In general, a site with an isosurface of Fukui function indicates its better reactivity. Fukui function is defined as

$$f(\vec{r}) = \left[\frac{\delta\mu}{\delta\upsilon(\vec{r})}\right]_N = \left[\frac{\partial n(\vec{r})}{\partial N}\right]_{\upsilon(\vec{r})} \tag{36}$$

$\mu$ is the electronic chemical potential $\upsilon(\vec{r})$ is the external potential, $N$ is the number of electrons in the present system, and $n(\vec{r})$ is the electron density. In practice, the donor Fukui function can be evaluated in its finite difference form:

$$f^-(\vec{r}) = n_N(\vec{r}) - n_{N-1}(\vec{r}) \tag{37}$$

$n_N$ and $n_{N-1}$ represent the electron densities of the system with $N$ electrons and $N$-1 electrons with identical structures, respectively. The donor Fukui function $f^-(\vec{r})$ shows the change of electron density at each position when the system is attacked by electrophilic reagents and then donates charge to electrophiles, reflecting the ability of electrons in different regions to participate in electrophilic reactions.

The donor Fukui functions of RhN$_4$, VN$_4$, Fe-Cu-N$_6$, FeN$_4$ and CuN$_4$ are shown in Fig. S27. The isosurface near the Fe site of FeN$_4$ is broader than that of Cu in CuN$_4$. Thus, the electrophilic reactivity of the Fe sites is much higher than the Cu sites (Fig. S27c–e). In the Fe-Cu-N$_6$ system, the two dual centers of Fe and Cu atoms can enhance their common neighboring N atoms' electrophilic reactivity. Therefore, the Fukui function suggests that the two-center N atoms may also contribute to catalytic processes (Fig. S27e). For RhN$_4$ and VN$_4$, the active region around Rh is wider than V, corresponding to Rh's higher electrophilic reactivity.

## Formation energy

To evaluate the stability of catalysts, we investigated the formation energy with the Vienna ab initio Simulation Package (VASP)[69,70]. The elemental core and valence electrons were represented using the projector augmented wave (PAW) method[71]. The Perdew–Burke–Ernzerhof generalized gradient approximation (GGA-PBE)[72] functional was employed to estimate the exchange–correlation potential energy. For simulation of five different Fe-Cu-N$_x$ bimetallic structures, a 12.78 Å × 12.30 Å supercell containing 58 atoms is sufficient. To avoid the interaction between layers due to periodicity and improve the reliability and accuracy of calculations, a 15 Å vacuum distance was taken along the z-direction perpendicular to the surface of catalysts. We ensured the convergence of calculations by setting the cutoff energy for the plane-wave basis at 500 eV. In addition, all Brillouin zone integrations were performed according to the Monkhorst Pack scheme[5], and $4 \times 4 \times 1$ k-mesh was used for relaxation calculations. For structural relaxation, the convergence criteria for energy and force were set to $1 \times 10^{-5}$ eV and $-0.01$ eV/Å, respectively. In this work, the formation energies ($E_f$) of different Fe-Cu-N$_x$ bimetallic structures were calculated by the following equation:

$$E_f = E_{FeCuN_xC} - E_{N_xC} - E_{Fe} - E_{Cu} \tag{38}$$

Where $E_{FeCuN_xC}$ and $E_{N_xC}$ are the total energies of Fe-Cu-N$_x$ systems and N$_x$C pore substrate, respectively. $E_{Fe}$ and $E_{Cu}$ are energy per atom for the Fe and Cu atoms, respectively.

## Animal experiment

All animal procedures were approved by the Institute of Radiation Medicine, Chinese Academy of Medical Science, and Peking Union Medical College, following the ethics and ethics rules of the Animal Committee and complying with the principles of animal protection, animal welfare, and ethics (IRM-DWLI-2021107). The animal models were established in Tianjin Medical University General Hospital.

## Fluid percussion injury (FPI) models

Male C57BL/6 J mice (6–8 weeks) were purchased from Beijing Vital River Laboratory Animal Technology Co., Ltd. and reared for 1 week in a constant temperature and humidity condition to adapt to the environment. Mice were housed in a constant temperature (21–23 °C) and animal humidity (45–60%) environment on a 12-h light-dark cycle, with food and water available ad libitum. Fluid percussion injury (FPI) models were conducted on male C57BL/6 J mice (7–9 weeks). Mice were intraperitoneally injected with 10% chloral hydrate (10 mg/kg), and the anesthesia state was judged through skin pinching reaction and toe stimulation reaction. Mice were fixed in a stereotaxic frame and depilated, and the scalp was cut with a 1.5–2 cm length wound. Craniotomy was performed by drilling the skull on the right side of the parietal-temporal cortex in a circle of 4 mm. FPI was established with a hydraulic impact operated at 1.9 atm. The scalp of injured mice was sutured with PGA sutures modified with MN$_x$ randomly and divided into RhN$_4$, VN$_4$, FeN$_4$, CuN$_4$, and Fe-Cu-N$_6$ groups as the administration groups ($n = 17$ per group). Notably, the MN$_x$-sutures were treated with a high-temperature sterilization procedure before use, and the entire animal experiments were conducted under a specific pathogen-free (SPF) environment. TBI groups were sutured with pure PGA suture with FPI as the vehicle group. The control groups were only depilated as wild animals without FPI. All mice were marked, classified, and put into cages under the SPF level environment. The scalp injury and body weight were tracked throughout the experiments.

## Histological analysis

Mice were anesthetized, and all organs were taken out on day 3 or day 4 post-injury. Scalp and brain tissues were fixed in neutral formalin solution for 48 h, embedded in paraffin, and mounted on slides (2 μm). Tissues were deparaffinized twice with xylene, dehydrated with gradient alcohol (100%, 95%, 80%, and 70%) for 5 minutes every time, and washed with phosphate-buffered saline (PBS) twice. Then, the slices were covered with a 3% $H_2O_2$ solution and kept at 37 °C for 15 minutes to inactivate endogenous peroxidase. Subsequently, the slides were immersed in a dye vat filled with a citrate buffer solution at 95 °C for

10 min to repair the antigen. After cooling naturally, slices were blocked with 10% goat serum albumin for 40 min at 37 °C. For immunofluorescence staining, the primary antibody was dropped onto the slices and incubated at 4 °C overnight. The primary antibody information is as followed: CD31 Polyclonal antibody (1:100, Proteintech, 28083-1-AP), CD68 Polyclonal antibody (1:100, Proteintech, 28058-1-AP), anti-TNFα antibody (1:100, Abcam, ab183218), anti-IL-1β antibody (1:100, Affinity, AF5103), anti-Iba1 antibody (1:300, Abcam, ab48004), and anti-NeuN antibody (1:100, Proteintech, 26975-1-AP). After the primary antibody was cleaned, CoraLite488-conjugated Affinipure Goat Anti-Rabbit IgG (H+L) (1:100, Proteintech, SA00006-2) was utilized as the secondary antibody in the dark for 1–1.5 h. Then slices were mounted with an anti-fade mounting medium with DAPI (S2110, Solarbio, China) and photographed with a fluorescence microscope (EVOS, AMG). Granulation tissue was examined by staining scalp sections for 3 days post-injury with hematoxylin-eosin (H&E) staining.

## Kit detection

Mice were anesthetized, and all organs were taken out and washed once in neutral PBS buffer on day 3 post-injury ($n = 3$ per group). Homogenates of scalp and brain tissues were centrifuged at 10,000 g for 10 min, and the supernatant was stored at −80 °C for preparation. An Enhanced BCA protein assay kit (Beyotime, P0010) was used to detect the protein concentration. Typical oxidative stress-related factors MDA, SOD, GSH/GSSG, and $H_2O_2$ for scalp and brain tissues were detected with a hydrogen peroxide assay kit (Beyotime, S0038), total SOD assay kit with WST-8 (Beyotime, S0101M), GSH and GSSG assay kit (Beyotime, S0053), and lipid peroxidation MDA assay kit (Beyotime, S0131S). In addition, inflammatory cytokines for scalp and brain tissues along with VEGF for scalp tissues were detected with IL-6 (Abcam, ab100713), IL-1β (Abcam, ab197742), TNF-α (Abcam, ab208348), and VEGF (Abcam, ab209882). These assays were conducted following the instructions provided by the manufacturers. GraphPad software was used for plotting.

## Flow cytometry

To evaluate the changes of immune cells in the wound and blood, we collected the scalp and blood in different groups on day 6 post brain injuries. The fresh wound tissues were digested with collagenase and hyaluronidase (Stemcell, 07919) overnight at 37 °C to obtain single cells for flow cytometry analysis (BD FACSCanto II). Lymphocytes in the blood were collected with Ficoll kits (Solarbio, P8620). Lymphocytes were stained with PE anti-mouse F4/80 (Biolegend, 123110), APC anti-mouse CD80 (Biolegend, 104714), FITC anti-mouse CD206 (MMR) (Biolegend, 141704), APC/Cyanine7 anti-mouse CD3 (Biolegend, 100222), PE/Cyanine7 anti-mouse CD8a (Biolegend, 100722), FITC anti-mouse CD4 (Biolegend, 100406), PE anti-mouse NK-1.1 (Biolegend, 156504), PerCp anti-mouse CD45 (Biolegend, 103130), APC anti-mouse CD19 (Biolegend, 152410), APC anti-mouse CD25 (Biolegend, 101909), and PE-Foxp3 (FJK-16s) (Thermo, 12-5773-82) antibodies, PE Rat IgG2a, κ Isotype Ctrl Antibody (Biolegend, 400508), FOXP3 Monoclonal Antibody (FJK-16s), PE (eBioscience™ Invitrogen™, 12-5773-82).

## Statistic methods

Data are presented as mean ± standard deviation (SD) or mean ± standard error of the mean. For multiple comparisons, the one-way analysis of variance (ANOVA) with one-sided Tukey's multiple comparisons test was used to assess the difference in means among groups.

## Reporting summary

Further information on research design is available in the Nature Research Reporting Summary linked to this article.

## Data availability

All data supporting the findings of this study are available within this article and Supplementary Information files. Source data are provided with this paper. Data is available from the corresponding authors upon request. Source data are provided with this paper.

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

## Acknowledgements
X.-D. Z. is financially supported by the National Natural Science Foundation of China (Grant Nos. 91859101, 81971744, U1932107, 82001952, and 11804248), The National Key Research and Development Program of China (2021YFF1200700), Outstanding Youth Funds of Tianjin (2021FJ-0009), National Natural Science Foundation of Tianjin (Nos. 19JCZDJC34000 and 20JCYBJC00940), the Innovation Foundation of Tianjin University, and CAS Interdisciplinary Innovation Team (JCTD-2020-08).

## Author contributions
X.-D.Z. conceived and designed the experiments. S.Z. contributed to the synthesis, structural characterization of the materials. S.Z., M.J., and X.L. contributed to the related activity measurement. Y.L. and L.L. contributed to the simulation of the theoretical calculation. S.S., S.Z., X.C., K.C., and H.M. participated in the biological experiments and data collection. S.L., J.Z., and T.L. provided technical supports during the research. X.-D.Z., S.Z., Y.L., S.S., X.M., L.L., J.Y., and H.W. analyzed the data and prepared the manuscript. All authors discussed the results and commented the manuscript.

## Competing interests
The authors declare no competing interests.
