## [Peer Review File · Nature Communications]

Single-Atom Nanozymes Catalytically Surpassing Naturally-occurring Enzymes as Sustained Stitching for Brain TraumaReviewers' comments:

Reviewer #1 (Remarks to the Author):

In this study, the authors developed a series of single-atom artificial enzymes RhN₄, VN₄, and Fe-Cu-N₆ and used them in scalp wound sutures. These materials show excellent POD, CAT, SOD and GPx and other enzyme-like catalytic activities as well as good antioxidant and anti-inflammatory effects. However, there have been many reports about nanozyme materials with multiple antioxidant capabilities. The material does not show obvious innovation in the synthesis method and overall design. Many key questions remain to be answered. Therefore, I cannot recommend the acceptance of this work in Nature Communications. Following suggestions may help the authors further revise the manuscript.

1. From Figure S1, the UV-vis spectra of all MN_x nanomaterials are almost a horizontal straight line without any characteristic absorption peaks. Please specify the reason.
2. I have not seen any data to support the Fe-Cu-N₆ bimetallic contiguous sites. Why can't Fe and Cu be MN₄ sites that exist in isolation from each other in the catalyst?
3. The single-atom catalysts involved in the research are all synthesized using ZIF-8 or ZIF-N/C structures as precursors, which contain a considerable amount of Zn, and the XPS characterization of the single-atom materials also measured the valence signal of Zn. So why did the authors not take Zn into account in the physical and chemical characterization and activity evaluation of these catalysts?
4. The authors compared the POD kinetic parameters of MN_x and HRP against TMB. However, in biological applications, since the main substrate in the lesion is H₂O₂, more attention should be paid to the POD kinetic parameters of the catalyst against H₂O₂.
5. Nanozymes reported in the past often exert POD-like catalytic activity by catalyzing H₂O₂ to generate hydroxyl radicals. In this study, can MN_x also catalyze H₂O₂ to produce hydroxyl radicals? The authors need to use ESR experiments to detect the changes in the content of free radicals in POD, CAT, SOD and other catalytic processes, and based on this to prove the reliability of DFT.
6. In previous reports, POD nanozymes have the effects of promoting oxidation and cell killing because they catalyze H₂O₂ to produce hydroxyl free radicals. It seems that the excellent POD activity of MN_x is somewhat contrary to its antioxidant effect. The author needs to explain this contradiction in depth.
7. The single-atom nanozymes used in this study and other reported single-atom nanozymes are all synthesized by the method of precursor pyrolysis. Why does the single-atom material synthesized in this study have more superior activity than other similar single-atom materials? Compared with previous reported materials, what are the advantages and innovations of MN_x in synthesis method and structure-activity characteristics?
8. The diagrams of the reaction pathways are not very clear visually, and higher resolution images are required.
9. The inflammatory response at the wound site has the effect of preventing microbial infection. Has the author considered whether MN_x may increase the possibility of wound infection after being anti-inflammatory and antioxidant?

Reviewer #2 (Remarks to the Author):

Minor

In this manuscript, through a unique bilateral reaction mechanism that decreases reaction energy barriers, Zhang et al constructed a series of single-atom artificial enzymes RhN₄, VN₄, and Fe-Cu-N₆, showing potent catalytic activities beyond natural enzymes. The RhN₄, VN₄, and Fe-Cu-N₆ displayed high peroxidase-like (POD-like) activity, catalase-like (CAT-like) activity, superoxide dismutase-like (SOD-like) activity, and glutathione peroxidase-like (GPx-like) activity. Moreover, surgical sutures based on RhN₄ and VN₄ showed recyclable catalytic features without apparent decay in 1 month and accelerated scalp healing from brain trauma via promoting the vascular endothelial growth factor (VEGF) and diminishing oxidative stress and inflammation. The authors presented a novel yet cost-effective approach to construct a kind of efficient and stable

artificial enzyme and successfully used RhN4, VN4, and Fe-Cu-N6 in the field of healing traumatic brain injury. The overall study was well organized and performed and the goals of this research were of particular interest in the field of nano-catalysis and bioengineering. The specific issues are suggested to be addressed as following:

- 1) The immune cells like macrophages, Tregs, natural killer cell, B cells as well as T cells play an important role in wound repairing. So, I strongly suggest the authors to analyze the changes of immune cells after various treatment via flow cytometry.
- 2) I have concerned that the metal on the MNx-based sutures could spread and release into the blood, which might lead to poisoning. I strongly suggest authors to detect the concentration of metal in the blood after various treatment.
- 3) It would be better to add important quantitative data in the abstract to highlight the significance of this manuscript.

Minor issues:

- 1) In Page 5, Line 99 in the main text, "(Figure 1j)" should be "(Figure 2j)".
- 2) Some references about wound repairing such as Matter, 2021,4 (9), 2985-3000, Small Structures, 2021, 2(7), 2100002 are suggested to be added.

Our Replies to the Reviewer' Comments and the Revisions Made to the Manuscript

Comments by Reviewer #1

In this study, the authors developed a series of single-atom artificial enzymes RhN₄, VN₄, and Fe-Cu-N₆ and used them in scalp wound sutures. These materials show excellent POD, CAT, SOD and GPx and other enzyme-like catalytic activities as well as good antioxidant and anti-inflammatory effects. However, there have been many reports about nanozyme materials with multiple antioxidant capabilities. The material does not show obvious innovation in the synthesis method and overall design. Many key questions remain to be answered. Following suggestions may help the authors further revise the manuscript.

Reply: Thank you very much for the review and comments. Indeed, our work doesn't focus on the synthesis of the materials but on filling and refining the **exploration of the active site, enzyme-like catalytic process, new catalytic mechanism, and inflammation regulated wound healing** of single-atom catalysts. Because of the great interest in single-atom catalysts with M-N₄ coordination structures, many MN₄-related works can be reported in prestigious journals, such as FeN₄, CoN₄, InN₄, NiN₄, FeN₃P, etc. (*Angew. Chem. Int. Ed.* 2019, 58, 18971-18980; *Angew. Chem. Int. Ed.* 2020, 59, 5108-5115; *Nat. Chem.* 2020, 12, 764-772; *Angew. Chem. Int. Ed.* 2020, 59, 3033 -3037; *Nat. Catal.* 2021, 4, 407-417; *Nat. Catal.* 2018, 1, 689-695; *Nat. Commun.* 2020, 11, 357). However, in our work, the key innovations are the following:

(1) **The discovery of active sites in RhN₄ and VN₄ with activity beyond natural enzymes.** The enzyme-mimic activity of FeN₄ and CuN₄ was reported, but RhN₄ and VN₄ were **NOT** presented in previous work. Our work firstly confirmed the high POD-, CAT-, GPx-like activity of RhN₄ and VN₄ with atomic level selectivity.

(2) **The unique catalytic mechanism and active unit of RhN₄ and VN₄.** Different from previous work, we found that RhN₄ and VN₄ showed the bilateral reaction mechanism, and discovered the active unit of Rh/V-O-N₄.

(3) **Inflammation regulated wound healing.** Traditional tail veins are associated with potential safety issues in biomedicine, we designed the highly catalytic sutures with long-term, recyclable properties, and achieve the wound healing via inflammation regulated mechanism, greatly reducing side effects.

We have revised the abstract and the introduction sections to highlight the innovative aspects of this work.

Changed in the revised manuscript (Pages 2-4):

“ABSTRACT. Regenerable artificial enzyme mimetics with high catalytic stability and sustainability are promising as a substitute for naturally-occurring enzymes to some extent. However, they are usually limited by insufficient catalytic activities and non-specific selectivity. Herein, we developed single-atom artificial enzymes of RhN₄, VN₄, and Fe-Cu-N₆ with extraordinary catalytic activities surpassing natural enzymes. Notably, Rh/VN₄ preferably formed an Rh/V-O-N₄ active center with circular catalytic processes via decreasing reaction energy barriers by 0.2 eV and showed 4 and 5-fold higher affinities in the peroxidase-like (POD-like) activity than FeN₄ and the natural horseradish peroxidase, respectively. Furthermore, RhN₄ presented an over 20-fold improved affinity in the catalase-like (CAT-like) activity compared to the natural CAT; Fe-Cu-N₆ displayed high selectivity towards the superoxide dismutase-like (SOD-like) activity; VN₄ favored a 7-fold higher glutathione peroxidase-like (GPx-like) activity than the natural GPx. The Rh/VN₄ mediated a “two-sided oxygen-linked” catalytic reaction path, resulting in high-performance biocatalytic activities in a mechanism distinct from the previously-reported single-atom catalysts. Bioactive sutures developed using RhN₄ and VN₄ showed recyclable catalytic features without apparent decay in one month and accelerated scalp healing from brain trauma by promoting the vascular endothelial growth factor (VEGF), regulating the immune cells like macrophages, and diminishing oxidative stress and inflammation.”

“Introduction

Artificial enzyme engineering with sustainabilities emerges as a versatile methodology for creating biocatalysts¹⁻⁶. The intrinsic feature of continuous electron transfers allows artificial enzymes to maintain high catalytic stability and arbitrarily tailor their characteristics⁷⁻¹². The superior catalytic activity over the naturally-occurring enzymes is one of the major challenges existing in designing artificial enzymes¹³⁻¹⁶. A feasible strategy is

to construct an artificial active center by imitating the biologically active center of enzymes or proteins^{2,17-22}. State-of-art artificial enzymes with an M-N₄ center inspired by cytochrome P450 have superior reaction rates and high substrate affinities close to the natural horseradish peroxidase (HRP)²³⁻²⁵. Expanded Fe-N₅ and Fe-N₃-P were reported to have 17- and 2-fold higher catalytic efficiencies than Fe-N₄^{14,26}, indicating their great potential in boosting bioactivities. However, the development of artificial enzymes with properties **outperforming** natural enzymes **remains unfulfilled and highly challenging**.

Other **unresolved and high-profile** questions are the underlying reaction mechanisms and detailed atomic coordination structures during biocatalytic reactions to endow catalytic selectivity^{13,27-29}. Unlike the well-defined structures of **single-atom catalysts**^{16,30,31}, traditional nanostructures without elucidated atomic structures on the surface cannot provide sufficient structure-function correlation to study the electron transfers at atomic levels³². For example, Pt and Fe₃O₄ exhibit peroxidase-like (POD-like) activities^{6,33}; Au and V₂O₅ are more selective for the glutathione peroxidase-like (GPx-like) activity³⁴⁻³⁶; Cu demonstrates a superior superoxide dismutase-like (SOD-like) activity³⁷. However, the selectivity of these biocatalysts **merely** relies on the specific atomic coordination and surface ligands of individual nanosystems, making it difficult to corroborate a universal catalytic mechanism explicitly. Besides, the complexity of reaction pathways also compromises catalytic reactions of artificial enzymes²⁹, leaving the atomic configuration and bond morphology inadequately clarified during the dynamic reaction process.

This work reports structurally well-defined and atomically-precise biocatalysts of RhN₄ and VN₄ **with high POD- and catalase (CAT)-like activities**, surpassing corresponding natural enzymes and reported FeN₄³⁸⁻⁴⁰. **Moreover, the RhN₄ and VN₄ favorably form the Rh/V-O-N₄ structure as the active center of circular catalytic processes to decrease reaction energy barriers significantly, which is fundamentally different from FeN₄^{14,41-43} (Figure 1). In particular, the catalytic mechanism of RhN₄ and VN₄ modulates a unique “two-sided oxygen-linked” catalytic reaction path, resulting in high catalytic efficiency. Meanwhile, the specific SOD activity was derived from Fe-Cu-N₆, and the GPx-like activity of VN₄ considerably exceeded that of the natural GPx. Furthermore, the high catalytic stability and recyclability enabled these artificial enzymes to be used as medical sutures for sustained scalp healing from brain trauma via a serial biocatalytic process. RhN₄ and VN₄ in sutures play an essential role in the proactive regulation and immune control of macrophages and vascular endothelial growth factors (VEGF).”**

1. From Figure S1, the UV-vis spectra of all MN_x nanomaterials are almost a horizontal straight line without any characteristic absorption peaks. Please specify the reason.

Reply: Thank you very much for the careful review. The MN_x nanomaterials were obtained by high-temperature pyrolysis of zinc-imidazole skeleton (ZIF-8). The main components are N-doped carbon (ZIF-N/C) and trace amounts of metal atoms in single-atom form. Based on the Mie and Maxwell-Garnett theories, it is clear that metals in single atomic form do not show absorption peaks, and the surface plasmon resonance (SPR) absorption peaks appear only at the particle levels. Therefore, the absorption spectra of horizontal straight lines demonstrate that the loaded metal atoms do not aggregate into metal oxides or metal particles, but exist as individual atoms, in agreement with the XRD and Raman results. Furthermore, according to previous reports in authoritative journals, the absorption spectra provided are all metal salt-containing precursors or ZIF-8, which are consistent with our results. No absorption characteristic peaks have been reported for single-atom catalysts, similar to our work (*Nat. Chem.* 2020, 12, 764–772; *Nat. Nanotechnol.* 2020, 15, 390–397; *Angew. Chem. Int. Ed.* 2020, 59, 5108–5115; *J. Am. Chem. Soc.* 2019, 141, 12005–12010).

2. I have not seen any data to support the Fe-Cu-N₆ bimetallic contiguous sites. Why can't Fe and Cu be MN₄ sites that exist in isolation from each other in the catalyst?

Reply: Thank you very much for this valuable recommendation. The specific reasons why the structure of Fe-Cu-N₆ was finally determined are mainly as follows. **Experimentally**, the Fe-Cu diatomic pairs could be obviously observed at the enlarged magnified AC-HAADF-STEM image of Fe-Cu-N₆ (**Figure S1a**). A characteristic distance between atomic pairs can be obtained via intensity profiles, where the average distance of Fe-Cu bimetallic pair is ~2.27 Å (**Figure S1b**). Crucially, the XANES results of Fe-Cu-N₆ show that the coordination number between the Fe or Cu atoms and the surrounding N atoms are both 4, and the bond lengths of Fe-N and Cu-N are the same as those of the monometallic FeN₄ and CuN₄. Moreover, the ratio of Fe and Cu content was measured to be about 1:1

by inductively coupled plasma-mass spectrometry (ICP-MS). Based on the above experimental results, there are five most likely structures, which are Fe-Cu-N₆, Fe-Cu-N₈-I, Fe-Cu-N₈-II, Fe-Cu-N₈-III, Fe-Cu-N₈-IV (**Figure S14**).

Figure S1. **a**, The magnified AC-HAADF-STEM images of Fe-Cu-N₆. **b**, The corresponding intensity distributions and the statistical Fe-Cu distance of the observed diatomic pairs.

Theoretically, the above five structures of Fe-Cu-N_x are finely predicted thermodynamically and kinetically using density functional theory (DFT) to verify the specific structure of Fe-Cu-N_x. It is well known that the lower the formation energy, the more stable the structure (*J. Phys. Chem. Lett.* 2021, 1, 2130-2145; *ChemElectroChem* 2020, 7, 3116-3122; *Catal. Sci. Technol.* 2018, 8, 996-1001). As shown in **Figure S14**, each metal atom is coordinated with four nitrogen atoms in Fe-Cu-N₆, Fe-Cu-N₈-I, Fe-Cu-N₈-II, Fe-Cu-N₈-III and Fe-Cu-N₈-IV structures, respectively. The isolated non-bonding Fe and Cu atoms locate in the four types of Fe-Cu-N₈ structures. The formation energy of these five different Fe-Cu-N_x bimetallic structures are calculated to further evaluate the stability of catalysts (**Table S3**). It is found that Fe-Cu-N₆ is the most thermodynamically favorable structure, due to the lowest formation energy (-5.432 eV) among the five structures. Moreover, the simulated distance of Fe-Cu in five different Fe-Cu-N_x structures has also been given out in **Table S3**. The simulated distance of Fe-Cu-N₆ is 2.46 Å, which most closely matches the experimental results (2.27 Å).

Figure S14. The optimized structures of five different Fe-Cu-N_x structures.

Table S3. The formation energies and the distance of Fe-Cu in five different Fe-Cu-N_x structures.

Structures	Fe-Cu-N ₆	Fe-Cu-N ₈ -I	Fe-Cu-N ₈ -II	Fe-Cu-N ₈ -III	Fe-Cu-N ₈ -IV
Formation energy (eV)	-5.432	-3.559	-4.285	-4.164	-2.967
Distance of Fe-Cu (Å)	2.46	5.04	4.36	4.86	4.04

In addition, detailed reaction pathways of enzyme-like of these five different Fe-Cu-N_x bimetallic structures have been calculated, for example POD-like processes. As shown in **Figure 3g** and **S22**, the energy barrier of Fe-Cu-N₆ structure is the lowest both in the POD (0.479 eV) processes. The other four structures reveal higher energy barriers in the POD processes (**Table S6**), and their activity order is inconsistent with the experimental observations that Fe-Cu-N_x possesses relatively high POD-like capacity. Therefore, combined with experimental and theoretical results, the Fe-Cu-N₆ is the most kinetically favorable structure among the five Fe-Cu-N_x bimetallic structures. In view of your comments, the detailed calculation results and corresponding discussion have been added in the revised manuscript.

Figure S22. POD processes with (a) Fe-Cu-N₈-I, (b) Fe-Cu-N₈-II, (c) Fe-Cu-N₈-III and (d) Fe-Cu-N₈-IV.

Table S6. The energy barriers of POD processes for five different Fe-Cu-N_x bimetallic structures.

Structures	Fe-Cu-N ₆	Fe-Cu-N ₈ -I	Fe-Cu-N ₈ -II	Fe-Cu-N ₈ -III	Fe-Cu-N ₈ -IV
Energy barrier	0.479	1.020	1.121	1.691	1.824

Added in the revised manuscript (Lines 1-4, Page 5):

“The Fe-Cu diatomic pairs could be observed in the enlarged magnified AC-HAADF-STEM image of Fe-Cu-N₆ (Figure S1a). The average distances between Fe and Cu atoms were calculated as ~2.27 Å from the statistical intensity profiles (Figure S1b).”

Added in the revised manuscript (Lines 1-8, Page 6):

“To characterize the exact Fe-Cu coordination, we thermodynamically and kinetically predicted five possible structures of Fe-Cu-N_x based on the above findings, defined as Fe-Cu-N₆, Fe-Cu-N₈-I, Fe-Cu-N₈-II, Fe-Cu-N₈-III, and Fe-Cu-N₈-IV (**Figure S14**). The simulated distance of Fe-Cu-N₆ was 2.46 Å (**Table S3**), which closely matched the experimental result of 2.27 Å. Fe-Cu-N₆ was identified as the most thermodynamically favorable structure due to the lowest formation energy of -5.432 eV by the Density Functional Theory (DFT) simulation (**Table S3**). Therefore, all of the subsequent calculations of the enzyme-like pathways were based on the Fe-Cu-N₆ structure.”

Added in the revised manuscript (Lines 17-19, Page 8):

“In addition, the simulated POD pathways for the five possible structures of Fe-Cu-N_x showed that Fe-Cu-N₆ structure possesses the lowest energy barrier (**Figure S22** and **Table S6**), further demonstrating Fe-Cu-N₆ is the most possible structure.”

Added in the revised manuscript (Lines 7-22 and 1-2, Pages 31-32):

“Formation energy

To evaluate the stability of catalysts, we investigated the formation energy with the Vienna ab initio Simulation Package (VASP)^{65,66}. The elemental core and valence electrons were represented using the projector augmented wave (PAW) method⁶⁷. The Perdew–Burke–Ernzerhof generalized gradient approximation (GGA-PBE)⁶⁸ functional was employed to estimate the exchange–correlation potential energy. For simulation of 5 different Fe-Cu-N_x bimetallic structures, a 12.78 Å × 12.30 Å supercell containing 58 atoms is sufficient. To avoid the interaction between layers due to periodicity and improve the reliability and accuracy of calculations, a 15 Å vacuum distance was taken along the z-direction perpendicular to the surface of catalysts. We ensured the convergence of calculations by setting the cutoff energy for the plane-wave basis at 500 eV. In addition, all Brillouin zone integrations were performed according to the Monkhorst Pack scheme⁵, and 4 × 4 × 1 k-mesh was used for relaxation calculations. For structural relaxation, the convergence criteria for energy and force were set to 1 × 10⁻⁵ eV and -0.01 eV/Å, respectively. In this work, the formation energies (E_f) of different Fe-Cu-N_x bimetallic structures were calculated by the following equation:

$$E_f = E_{FeCuN_xC} - E_{N_xC} - E_{Fe} - E_{Cu} \quad (38)$$

Where E_{FeCuN_xC} and E_{N_xC} are the total energies of Fe-Cu-N_x systems and N_xC pore substrate, respectively. E_{Fe} and E_{Cu} are energy per atom for the Fe and Cu atoms, respectively.”

Added in the references of the revised manuscript (Page 39):

- 65 Kresse, G. & Furthmüller, J. Efficiency of ab-initio total energy calculations for metals and semiconductors using a plane-wave basis set. *Comput. Mater. Sci.* **6**, 15-50 (1996).
- 66 Georg Kresse, J. F. Efficient iterative schemes for ab initio total-energy calculations using a plane-wave basis set. *Phys. Rev., B Condens. Matter* **54**, 11169-11186 (1996).
- 67 Blöchl, P. E. Projector augmented-wave method. *Phys. Rev. B* **50**, 17953-17979 (1994).
- 68 Perdew, J. P. *et al.* Atoms, molecules, solids, and surfaces: Applications of the generalized gradient approximation for exchange and correlation. *Phys. Rev. B* **46**, 6671-6687 (1992).

Added in the revised supporting information (Figure S1, Page 2, Figure S14, Page 15 and Figure S22, Page 23):

Figure S1. **a**, The magnified AC-HAADF-STEM images of Fe-Cu-N₆. **b**, The corresponding intensity distributions and the statistical Fe-Cu distance of the observed diatomic pairs.

Figure S14. The optimized structures of five different Fe-Cu-N_x structures.

Figure S22. POD processes with (a) Fe-Cu-N₈-I, (b) Fe-Cu-N₈-II, (c) Fe-Cu-N₈-III and (d) Fe-Cu-N₈-IV.

Added in the revised supporting information (Table S3, Page 60 and Table S6, Page 63):

Table S3. The formation energies and the distance of Fe-Cu in five different Fe-Cu-N_x structures.

Structures	Fe-Cu-N ₆	Fe-Cu-N ₈ -I	Fe-Cu-N ₈ -II	Fe-Cu-N ₈ -III	Fe-Cu-N ₈ -IV
Formation energy (eV)	-5.432	-3.559	-4.285	-4.164	-2.967
Distance of Fe-Cu (Å)	2.46	5.04	4.36	4.86	4.04

Table S6. The energy barriers of POD processes for five different Fe-Cu-N_x bimetallic structures.

Structures	Fe-Cu-N ₆	Fe-Cu-N ₈ - I	Fe-Cu-N ₈ - II	Fe-Cu-N ₈ - III	Fe-Cu-N ₈ - IV
Energy barrier (eV)	0.479	1.020	1.121	1.691	1.824

3. *The single-atom catalysts involved in the research are all synthesized using ZIF-8 or ZIF-N/C structures as precursors, which contain a considerable amount of Zn, and the XPS characterization of the single-atom materials also measured the valence signal of Zn. So why did the authors not take Zn into account in the physical and chemical characterization and activity evaluation of these catalysts?*

Reply: Thank you very much for the useful suggestion. We have considered the effect of Zn atoms on the MN_x artificial enzyme in both characterization and performance tests. The distribution of Zn elements was confirmed in energy-dispersive X-ray spectroscopy (EDS) mapping (not showed in the original manuscript) and supplemented in **Figure S2**. The valence state of Zn atoms was measured by XPS (showed in **Figure S6-S8** of the original manuscript).

Figure S2. Energy-dispersive X-ray spectroscopy (EDS) mapping images of (a) RhN₄, (b) VN₄, and (c) Fe-Cu-N₆, respectively.

Furthermore, we evaluated the catalytic performance of ZIF-N/C with only Zn atoms anchored under the same concentration and test conditions, including POD, CAT, SOD and GPx-like activities. As shown in **Figure S33**, ZIF-N/C showed a negligible catalytic activity, especially SOD and GPx-like activities compared to the blank control, indicating that the presence of Zn atoms had little effect on the catalytic performance of MN_x. The results are also consistent with previous reports (*ACS Catal.* 2020, 10, 11, 6422-6429). According to your comments, we have supplemented the results of structural characterization and activity assessment in the Supplementary Information and added the corresponding discussion in the revised manuscript.

Figure S33. **a**, Absorbance-time curves of the TMB chromogenic reaction catalyzed by ZIF-N/C (3 ng/ μ L). **b**, The POD-like activity of ZIF-N/C. **c**, SOD-like activity with and without ZIF-N/C (30 ng/ μ L). **d**, Reaction-time curves of the decomposition of H_2O_2 catalyzed by ZIF-N/C (30 ng/ μ L). **e**, The corresponding H_2O_2 clearance rate of ZIF-N/C (30 ng/ μ L). **f**, The GPx-like activity with and without ZIF-N/C (30 ng/ μ L).

Changed in the revised manuscript (Lines 5-6, Page 5):

“Single metal atoms and C and N elements are uniformly distributed throughout the entire nanostructure (**Figures 2c, f, i, and S2**).”

Added in the revised manuscript (Lines 19-23, Page 12):

“Moreover, to exclude the interference from Zn atoms on the catalytic performance of MN_x , the multiple enzyme-like activities of ZIF-N/C only anchored by Zn atoms were also tested (**Figure S33**). The ZIF-N/C showed a negligible catalytic activity, validating that the ultrahigh catalytic properties predominantly originate from the $M-N_x$ active centers.”

Added in the revised supporting information (Figure S2, Page 3 and Figure S33, Page 34):

Figure S2. Energy-dispersive X-ray spectroscopy (EDS) mapping images of **(a)** RhN₄, **(b)** VN₄, and **(c)** Fe-Cu-N₆, respectively.

Figure S33. **a**, Absorbance-time curves of the TMB chromogenic reaction catalyzed by ZIF-N/C (3 ng/ μ L). **b**, The POD-like activity of ZIF-N/C. **c**, SOD-like activity with and without ZIF-N/C (30 ng/ μ L). **d**, Reaction-time curves of the decomposition of H_2O_2 catalyzed by ZIF-N/C (30 ng/ μ L). **e**, The corresponding H_2O_2 clearance rate of ZIF-N/C (30 ng/ μ L). **f**, The GPx-like activity with and without ZIF-N/C (30 ng/ μ L).

4. The authors compared the POD kinetic parameters of MN_x and HRP against TMB. However, in biological applications, since the main substrate in the lesion is H_2O_2 , more attention should be paid to the POD kinetic parameters of the catalyst against H_2O_2 .

Reply: Thank you very much for this comment. We have performed kinetics measurement with H_2O_2 as a substrate (showed in the **Figures S18** and **S19** of original manuscript) and analyzed the relevant parameters. Most previously reported Michaelis-Menten constants (K_m) of single-atom enzymes are in the range of 12-40 mM (*Anal. Chem.* 2019, 91, 11994-11999; *Nano Today* 2020, 35, 100971; *Biosens. Bioelectron.* 2019, 142, 111495; *Adv. Funct. Mater.* 2020, 30, 1905410), while RhN_4 and VN_4 in our work are 12.38 and 14.18 mM, respectively, indicating a good binding affinity for H_2O_2 (**Table R1**). The results further confirm the excellent catalytic performance. However, we investigated the reported single-atom catalysts and found that the affinity for H_2O_2 is still lower than that of HRP, which is also the direction that single-atom nanozymes need to be further improved for the catalytic selectivity. Based on your suggestions, we have also added the table of kinetic parameters and related discussions in the revised manuscript.

Table R1. Kinetic parameters of POD-like activity of single-atom enzymes with the substrate of H_2O_2 .

Single-atom enzymes	Active sites	Substrate	K_m (mM)	Reference
RhN_4	Rh- N_4	H_2O_2	12.38	This work
VN_4	V- N_4	H_2O_2	14.18	
Fe-N-C SAzyme	Fe- N_x	H_2O_2	12.2	Anal. Chem. 2019, 91, 11994-11999.
FeBNC	Fe- N_x	H_2O_2	25.45	Nano Today 2020, 35, 100971
FeNC	Fe- N_x	H_2O_2	24.25	Nano Today 2020, 35, 100971
Fe-N-C SAN	Fe- N_x	H_2O_2	28.30	Biosens. Bioelectron. 2019, 142, 111495

Added in the revised manuscript (Lines 1-5, Page 7):

“The kinetic parameters of MN_x for H₂O₂ substrates were also quantified (Figure S19, S20, and Table S5). Compared to the reported K_m values of single-atom catalysts^{38,40,43,52}, MN_x showed a superior binding affinity for H₂O₂, confirming the outstanding catalytic properties. However, all single-atom artificial enzymes exhibited lower binding affinities for H₂O₂ substrates than HRP, which remains an unconquered challenge that would need further improvements.”

Added in the revised supporting information (Table S5, Page 62):

Table S5 Comparison of the kinetic parameters between MN_x and HRP towards the H₂O₂ substrate during the POD-mimic catalysis.

Catalysts	[E] (M)	Substance	K_m (μM)	V_m ($\mu\text{M min}^{-1}$)	k_{cat} (min^{-1})
RhN ₄	5.61×10^{-8}	H ₂ O ₂	12.38	5.08	0.903×10^2
VN ₄	1.28×10^{-7}	H ₂ O ₂	14.18	7.92	0.619×10^2
FeN ₄	7.50×10^{-7}	H ₂ O ₂	12.45	15.05	0.201×10^2
CuN ₄	6.57×10^{-7}	H ₂ O ₂	18.55	3.84	0.058×10^2
Fe-Cu-N ₆	5.55×10^{-7}	H ₂ O ₂	14.47	21.15	0.381×10^2
HRP	2.00×10^{-10}	H ₂ O ₂	2.63	1.97	9.85×10^3

[E], the molar concentration of the active sites; K_m , the Michaelis constant; V_m , maximal reaction velocity; k_{cat} , catalytic constant, where $k_{cat} = V_m/[E]$.

Added in the references of the revised manuscript (Pages 37-38):

- 38 Jiao, L. *et al.* Boron-doped Fe-N-C single-atom nanozymes specifically boost peroxidase-like activity. *Nano Today* **35** (2020).
- 40 Jiao, L. *et al.* Fe-N-C single-atom nanozymes for the intracellular hydrogen peroxide detection. *Anal. Chem.* **91**, 11994-11999 (2019).
- 52 Niu, X. *et al.* Unprecedented peroxidase-mimicking activity of single-atom nanozyme with atomically dispersed Fe-N_x moieties hosted by MOF derived porous carbon. *Biosens. Bioelectron.* **142** (2019).

5. Nanozymes reported in the past often exert POD-like catalytic activity by catalyzing H₂O₂ to generate hydroxyl radicals. In this study, can MN_x also catalyze H₂O₂ to produce hydroxyl radicals? The authors need to use ESR experiments to detect the changes in the content of free radicals in POD, CAT, SOD and other catalytic processes, and based on this to prove the reliability of DFT.

Reply: Thank you very much for your valuable comment. For the POD- and CAT-mimicking reaction processes, which have been reported to produce hydroxyl radicals ($\cdot\text{OH}$), and whether as-developed MN_x can catalyze H₂O₂ to generate $\cdot\text{OH}$ was verified by ESR measurements. The results revealed that the $\cdot\text{OH}$ can be produced by FeN₄-catalyzed H₂O₂ under acidic conditions (pH=4.5) (Figure R1a), similar to reported work (*Angew. Chem. Int. Ed.* 2019, 131, 4965-4970; *Nat. Commun.* 2018, 9, 1440), while almost no $\cdot\text{OH}$ was produced under weakly acidic and neutral environment (pH=6.1 or 7.3). Figure R1b showed the FeN₄-catalyzed $\cdot\text{OH}$ radical peaks are relatively stronger than those of RhN₄ and VN₄, showing capability to produce $\cdot\text{OH}$. Previous reports also confirmed that the production of $\cdot\text{OH}$ was closely related to the amount of pyrrolic N in the single-atom enzymes (*Angew. Chem. Int. Ed.* 2019, 131, 4965-4970; *J. Am. Chem. Soc.* 2018, 140, 12469–12475; *ACS Nano* 2019, 13, 2643-2653; *Small* 2019,

15, 1901834; *Angew. Chem. Int. Ed.* 2020, 59, 14498–14503). In our work, the XPS spectra confirm that MN_x contain small amounts of pyrrolic N, which is responsible for the trace $\cdot OH$ production catalyzed by MN_x . On the other hand, the addition of MN_x to H_2O_2 leads to an immediate production of abundant oxygen, which has been photographed and shown in **Figure S25** (original manuscript).

Figure R1. Detection of $\cdot OH$ generation. **a**, DMPO spin-trapping ESR spectra of FeN_4 recorded under different conditions. **b**, ESR spectra of MN_x in the absence or presence of H_2O_2 under acidic conditions.

The CAT process simulated by the DFT in our work was also performed based on this experimental result. Therefore, a small amount of $\cdot OH$ generated during the catalytic process would be immediately scavenged and hardly detected by ESR spectra. As for the SOD-mimicking reaction, no free radicals were generated during the catalytic process. In our work, the experiment based on NBT color development reaction is the detection of scavenging superoxide anion, which is consistent with the DFT-simulated SOD pathways. Notably, the “*” marks in the **Figures 2-4** all represent MN_x artificial enzymes. In addition, simulation and exploration of catalytic mechanisms for nanomaterials based on density functional theory has been an effective and versatile approach favored by researchers for many years. The power of density functional theory in exploring atomic-level catalytic mechanisms and kinetics for catalysis has been demonstrated in previous studies. (*ACS Catal.* 2020, 10, 12657-12665; *Natl. Acad. Sci. U.S.A.* 2011, 108, 937; *J. Phys. Chem. C* 2008, 112, 1308-1311. 24,25) what’s more, the simulations in our work were carried out entirely on the basis of experimental conditions, and the calculated results can all be matched with experimental results, again confirming the very high reliability.

6. In previous reports, POD nanozymes have the effects of promoting oxidation and cell killing because they catalyze H_2O_2 to produce hydroxyl free radicals. It seems that the excellent POD activity of MN_x is somewhat contrary to its antioxidant effect. The author needs to explain this contradiction in depth.

Reply: Thank you very much for the useful comments. By investigating the existing work reported in specialized and authoritative journals, it was found that nanozymes with only POD or OXD activity are usually applied in antibacterial (*Sci. Adv.* 2019, 5, eaav5490; *Small* 2019, 15, e1901834) and tumor therapy (*ACS Nano* 2019, 13, 2643-2653; *ACS. Appl. Mater. Interfaces* 2019, 11, 35228-35237; *ACS. Appl. Mater. Interfaces* 2018, 10, 35327-35333), while nanozymes with multiple enzyme-like activities or ROS scavenging performance are often effective in the treatment of biological disease models related to antioxidant and oxidative stress, such as traumatic brain injury (TBI), wound healing, and sepsis (*Angew. Chem. Int. Ed.* 2020, 59, 5108-5115; *Angew. Chem. Int. Ed.* 2017, 129, 11557-11561; *ACS. Appl. Bio. Mater.* 2020,3, 1147-1157; *Chem. Commun.* 2019, 55, 14534-14537; *Chem. Commun.* 2018, 55, 159-162). In our work, MN_x with multiple enzyme-like activities was proved to be effective in accelerating wound healing at the animal level, which is consistent with previously reported results. What’s more, redox in vivo is a dynamic equilibrium process, and MN_x artificial enzyme hold multiple enzyme-mimicking activities, and thus the application of MN_x in biomedicine cannot be determined with respect to a single catalytic reaction. To decide whether an artificial enzyme possesses a pro-oxidant or antioxidant effect, it needs to be placed in an actual biological environment to judge its real efficacy.

Figure R1. Detection of $\cdot\text{OH}$ generation. **a**, DMPO spin-trapping ESR spectra of FeN_4 recorded under different conditions. **b**, ESR spectra of MN_x in the absence or presence of H_2O_2 under acidic conditions.

According to reviewer's comments, whether the MN_x catalyze H_2O_2 to produce $\cdot\text{OH}$ during the POD-mimicking reaction, which was validated by ESR test. **Figure R1a** showed that almost no $\cdot\text{OH}$ were generated by FeN_4 -catalyzed H_2O_2 under weakly acidic and neutral conditions, while the weak ESR signals were captured only in the acidic environment. The ESR spectra appear to be a superposition peak of $\cdot\text{OH}$ with other radicals, which is different from the previously reported (*Angew. Chem. Int. Ed.* 2021, 60, 3001–3007; *Nat. Commun.* 2018, 9, 1440). As shown in **Figure R1b**, the FeN_4 -catalyzed radical peaks are relatively stronger than those of RhN_4 and VN_4 . On the other hand, it has been reported that the production of $\cdot\text{OH}$ is closely related to the content of pyrrolic N in the single-atom catalysts (*Angew. Chem. Int. Ed.* 2019, 131, 4965–4970; *J. Am. Chem. Soc.* 2018, 140, 12469–12475; *ACS Nano* 2019, 13, 2643–2653; *Small* 2019, 15, 1901834; *Angew. Chem. Int. Ed.* 2020, 59, 14498–14503). In our work, the XPS spectra confirm that RhN_4 and VN_4 contain small amounts of pyrrolic N, which is responsible for the trace $\cdot\text{OH}$ production catalyzed by RhN_4 and VN_4 . In addition, we measured the variation of POD-like activity with pH and the POD-like activity was stronger only in acidic environment (pH = 4 and 5) (**Figure R2**), which was consistent with the trend of $\cdot\text{OH}$ production. The biomedical application of MN_x was in a wound model with inflammation, whose physiological environment is slightly acidic with a pH range of 6.0 to 6.5 (*J. Leukocyte Biol.* 2001, 69, 522–530; *Crit. Care* 2004, 8, 331–336; *PLoS Path.* 2013, 9, e1003282), in which $\cdot\text{OH}$ formation is not detected. Furthermore, we also investigated the scavenging ability of MN_x for $\cdot\text{OH}$ at pH=6.1. Under UV irradiation, $\cdot\text{OH}$ generated by H_2O_2 was captured by DMPO as a distinct ESR peak, and the signal disappeared immediately after the addition of 100 $\mu\text{g/mL}$ of MN_x , indicating that MN_x possesses outstanding $\cdot\text{OH}$ scavenging ability in the inflammatory environment (**Figure S24**). To sum up, experimental results demonstrate that MN_x has effective $\cdot\text{OH}$ scavenging performance and exhibits antioxidant effects.

In addition, we further investigated the POD-mimicking catalytic mechanism of MN_x by DFT calculations to explore whether $\cdot\text{OH}$ are produced. The $\cdot\text{OH}$ -producing pathway similar to most previously reported and non- $\cdot\text{OH}$ -producing pathway were calculated in detail. For FeN_4 , there is no significant difference in the energy barriers in the two reaction paths, so that $\cdot\text{OH}$ are produced at pH = 4.5 and are scavenged at pH = 6.1 (**Figure S23** and **Table S7**). However, RhN_4 and VN_4 showed significant difference in the energy barriers between the two reaction paths (**Figure S23** and **Table S7**), so that a trace amount of $\cdot\text{OH}$ is produced at pH = 4.5 and the ability to remove $\cdot\text{OH}$ is stronger at pH = 6.1. Therefore, for the POD-mimic process of MN_x , the reaction pathways with and without $\cdot\text{OH}$ generation are both possible, which are closely related to the pH of the catalytic environment, the MN_x artificial enzymes prefer to scavenge $\cdot\text{OH}$ and exhibit antioxidant effects. The corresponding results and discussion have been added in the revised manuscript.

Figure R2. PH-dependent POD-like activity of (a-b) RhN₄ and VN₄ at 0.3 μM, (c-e) FeN₄, CuN₄ and FeCu-N₆ at 1.0 μM.

Added in the revised manuscript (Lines 1-13, Page 9):

“It is worth noting that, unlike most previous studies^{23,41,42}, the POD-mimicking reaction pathway simulated in this work is not accompanied by the generation of hydroxyl radicals ([•]OH). Instead, the [•]OH-producing catalytic reaction pathways were still simulated. As shown in **Figure S23** and **Table S7**, there are no significant differences in the energy barriers of FeN₄ between the two paths at 0.539 eV and 0.441 eV, suggesting the existence of reaction pathways both with and without generated [•]OH. However, all the other catalysts except FeN₄ display significant differences in energy barriers between the two reaction paths, especially RhN₄ and VN₄, indicating the preference for the pathway without [•]OH generation. We utilized electron spin resonance (ESR) spectroscopy to verify further whether MN_x can scavenge [•]OH in the inflammatory physiological environment (pH = 6.0-6.5). It turned out that MN_x effectively scavenged [•]OH (**Figure S24**). Therefore, exceedingly different from the widely reported FeN₄, the catalysis of RhN₄ and VN₄ follows unique paths and mechanisms, forming the Rh/V-O-N₄ active center by preference and then activating the circulatory catalytic reaction cascade.”

Added in the revised manuscript (Lines 9-16, Page 25):

“[•]OH scavenging test

The [•]OH scavenging measurements were conducted on an ESR spectrometer (JES-FA200, JEOL, Japan). The nitrogen trap DMPO was used to trap [•]OH to form DMPO-OH spin adducts. During the assay, [•]OH was generated by UV illumination of a centrifuge tube containing DMPO (50 mM), H₂O₂ (2 mM), and acetate buffer (pH = 6.1) for about 20 minutes. Then, the mixture was transferred into a quartz tube for ESR measurement, and a clear 4-peak signal with an intensity of 1:2:2:1 was captured. Subsequently, MN_x (100 ng/μL) was added, and the ESR test was performed again immediately.”

Added in the references of the revised manuscript (Page 37):

- 41 Jiao, L. *et al.* Densely isolated FeN₄ sites for peroxidase mimicking. *ACS Catal.* **10**, 6422-6429 (2020).
- 42 Huo, M., Wang, L., Wang, Y., Chen, Y. & Shi, J. Nanocatalytic tumor therapy by single-atom catalysts. *ACS Nano* **13**, 2643–2653 (2019).

Added in the revised supporting information (Figure S23, Pages 24 and Figure S24, Pages 25):

Figure S23. POD processes for generating $\cdot\text{OH}$ with (a) RhN_4 , (b) VN_4 , (c) FeN_4 , (d) CuN_4 and (e) Fe-Cu-N_6 .

Figure S24. The $\cdot\text{OH}$ scavenging ability of MN_x at 100 ng/ μL using DMPO as the radical-scavenging nitrogen trap.

Added in the revised supporting information (Table S7, Page 64):

Table S7. The energy barriers of POD processes with and without generating $\cdot\text{OH}$ of MN_x .

Catalysts		RhN ₄	VN ₄	FeN ₄	CuN ₄	Fe-Cu-N ₆
Energy barrier (eV)	·OH-producing	1.120	3.427	0.441	2.472	1.284
	non-·OH-producing	0.213	0.226	0.539	1.044	0.479

7. *The single-atom nanozymes used in this study and other reported single-atom nanozymes are all synthesized by the method of precursor pyrolysis. Why does the single-atom material synthesized in this study have more superior activity than other similar single-atom materials? Compared with previous reported materials, what are the advantages and innovations of MN_x in synthesis method and structure-activity characteristics?*

Reply: Thank you very much for the careful review. Indeed, we used the similar method to synthesize the MN₄, but we introduced different metal atoms Rh and V during the synthesis process of precursor pyrolysis to form different active center units. **The enzyme-like activity of RhN₄ and VN₄, even catalytic properties were NOT reported before. The different electron arrangements of the central metal atom show different reductions in the energy barriers during the enzyme-mimicking reactions, resulting in a significant effect on the catalytic performance.** We have synthesized a series of single-atom artificial enzymes and found that RhN₄ and VN₄ possess better enzyme-like activities. The kinetic activity results of as-prepared FeN₄ are in general agreement with the previous reports, and the corresponding *K_m* values are shown in the following **Table R2**. The systematic study of the catalytic mechanism of MN_x by DFT revealed that the high catalytic properties of Rh/VN₄ were attributed to two aspects. **On the one hand**, Rh/VN₄ preferentially forms the Rh/V-O-N₄ structure as active center of circular catalytic processes to significantly decreases reaction energy barriers, shortens the reaction path and accelerates the catalytic rate, which is very different from the reported FeN₄ and CuN₄ (*Nat. Catal.* 2021, 4, 407-417; *ACS Catal.* 2020, 10, 11, 6422-6429; *ACS Catal.* 2020, 10, 12657-12665; *Adv. Funct. Mater.* 2020, 30, 1905410; *Nano Today* 2021, 41, 101317; *Chem. Commun.* 2019, 55, 5271-5274; *J. Name.* 2012, 00, 1-3). **On the other hand**, Rh/VN₄ carries out a unique bilateral reaction path to greatly improve the catalytic efficiency, compared to the previous one-side reaction route (*ACS Nano* 2019, 13, 2643-2653; *Chem. Commun.* 2019, 55, 5271-5274; *ACS Catal.* 2020, 10, 6422-6429). Therefore, the RhN₄ and VN₄ single-atom artificial enzyme exhibits superior activity than other similar single-atom materials.

In addition, most reports have focused on POD- and CAT-like activities (*Nat. Catal.* 2021, 4, 407-417; *ACS Catal.* 2020, 10, 6422-6429; *ACS Catal.* 2020, 10, 12657-12665; *Nano Today* 2021, 41, 101317), while relatively few studies have been conducted on SOD- and GPx-like activities. Crucially, the development of single-atom artificial enzymes with different metal-based active centers has achieved the major challenge of regulating catalytic selectivity at the atomic level. Furthermore, exceedingly different from reports without a catalytic mechanism or with only a single catalytic reaction pathway (*Angew. Chem. Int. Ed.* 2020, 59, 5108-5115; *Angew. Chem. Int. Ed.* 2020, 59, 14498-14503; *Nano Today* 2020, 35, 100971; *Nat. Catal.* 2021, 4, 407-417; *Sci. Adv.* 2019; 5, eaav5490; *ACS Catal.* 2020, 10, 6422-6429; *Adv. Funct. Mater.* 2020, 30, 1905410; *Chem. Commun.* 2019, 55, 5271-5274), we have systematically simulated multiple enzyme-mimic process of MN_x via DFT, including POD, CAT, SOD and GPx, refining the investigation of the catalytic mechanism for single-atom catalysts.

As a result, our work doesn't focus on the synthesis of the materials, but on the innovations of RhN₄ and VN₄ in catalytic mechanism (**new active unit of Rh/V-O-N₄**), catalytic path (**unique bilateral reaction path**) and multienzyme-like activities (**especially GPx-like activities**). Based on your comments, we have added the relevant discussion to illustrate the advantages and innovations of MN_x in terms of structure-activity characteristics at the appropriate locations of revised manuscript.

Table R2. Kinetic parameters of POD-like activity of single-atom enzymes with the substrate of TMB and H₂O₂.

Single-atom enzymes	Active sites	Substrate	K_m (mM)	V_m (M s ⁻¹)	Activities	Reference
--------------	-----------	---------------------------	---	------------	-----------

FeN₄	Fe-N ₄	TMB	0.184	2.06×10 ⁻⁷	POD, CAT	This work
		H ₂ O ₂	12.45	2.5×10 ⁻⁷		
Fe-N-C	Fe-N _x	TMB	0.230	1.33×10 ⁻⁷	OXD	Angew. Chem. Int. Ed. 2020, 59 14498
Fe SSN	Fe-N ₄	TMB	0.53	2.04×10 ⁻⁷	POD	Small 2020, 16, 2002343
		H ₂ O ₂	0.36	1.32×10 ⁻⁷		
Fe SAEs	Fe-N ₄	TMB	3.92	5.88×10 ⁻⁷	POD, OXD, CAT	Chem. Commun. 2019, 55, 2285-2288.
		H ₂ O ₂	0.243	8.25×10 ⁻⁸		
Fe-N-C SAzyme	Fe-N _x	TMB	3.6	1.16×10 ⁻⁶	POD	Anal. Chem. 2019, 91, 11994-11999.
		H ₂ O ₂	12.2	3.56×10 ⁻⁷		
FeBNC	Fe-N _x	TMB	2.02	2.22×10 ⁻⁶	POD	Nano Today 2020, 35, 100971
		H ₂ O ₂	25.45	1.38×10 ⁻⁶		
FeNC	Fe-N _x	TMB	1.48	0.56×10 ⁻⁶	POD	Nano Today 2020, 35, 100971
		H ₂ O ₂	24.25	5.48×10 ⁻⁷		
Fe-N-C SAN	Fe-N _x	TMB	0.08	7.45×10 ⁻⁷	POD, OXD	Biosens. Bioelectron. 2019, 142, 111495
		H ₂ O ₂	28.30	4.29×10 ⁻⁷		
Fe-NrGO	Fe-N ₄	TMB	0.074	1.74× 10 ⁻⁶	POD	Adv. Funct. Mater. 2020, 30, 1905410
		H ₂ O ₂	43	1.44× 10 ⁻⁶		
FeN₅ SA/CNF	FeN ₅ /C	TMB	0.148	7.58×10 ⁻⁷	OXD	Sci. Adv. 2019, 5, eaav5490,
Fe-N/C	Fe-N	TMB	0.205	5.411×10 ⁻⁷	OXD	Analyst, 2021, 146, 207

Changed in the revised manuscript (lines 9-23, Pages 8):

“The results reveal that the ultrahigh catalytic activity of MN_x is primarily attributed to the unique “two-sided oxygen-linked” catalytic reaction path to increase the utilization of catalysts compared to the previously-reported one-side reaction route^{41,42}. In the catalytic procedure of VN₄, the “initialized” state enrolls an extra hydrogen atom for attachment compared to RhN₄ and Fe-Cu-N₆. The finalization step, which restores catalysts to their original states, is only observed in specific reaction pathways, including the Fe-Cu-N₆ and FeN₄ processes, the finalization process, and the reinstated initialization step of subsequent catalysis, but not in the catalytic processes of the RhN₄ and VN₄ loop between bilateral adsorption and hydrogen adsorption steps (**Figure 3e-f**). Therefore, Rh/VN₄ preferentially forms the Rh/V-O-N₄ structure with an active center for the circular catalytic processes to significantly decrease reaction energy barriers, shorten the reaction path, and accelerate the catalytic kinetics, thus creating a unique bilateral reaction path that gives rise to a catalytic performance higher than the reported FeN₄ and CuN₄”

8. The diagrams of the reaction pathways are not very clear visually, and higher resolution images are required.

Reply: Thank you very much for your kind comment. Base on your suggestion, we have carefully adjusted the size of the POD and CAT reaction pathway diagrams .The corresponding figures have been changed in the revised manuscript.

Changed in the revised manuscript (Figures 3-4, Pages 44-45):

Figure 3. The POD-like activity of MN_x . **a**, Reaction-time curves of the TMB colorimetric reaction catalyzed by 1 μM MN_x , with the substrate concentration of TMB and H_2O_2 at 800 μM and 0.2 M. **b**, Quantification of specific POD-like activities of MN_x . The specific activity value ($\text{U}/\mu\text{mol}$) was determined by dividing the POD-like activity by the metal-based active sites. **c**, Comparison of the stability of POD-like activities between MN_x and natural enzymes. All tests are performed at room temperature. **d**, The energy barrier of MN_x in POD-mimic reaction pathway via DFT simulation. **e-f**, Schematic illustration of the cyclable catalytic POD processes for (e) Fe-Cu-N_6 and (f) Rh/V-N_4 . **g**, POD processes with MN_x (Rh, V, and Fe-Cu). White, dark grey, blue, red, cyan, light grey, orange, and brown balls represent H, C, N, O, Rh, V, Fe, and Cu atoms. The MN_x catalyst is represented by an asterisk (*) for clarity.

Figure 4. The CAT-like activity of MN_x. **a**, Reaction-time curves of the decomposition of H₂O₂ catalyzed by 4.8 μM MN_x. **b**, Quantification of specific CAT-like activities of MN_x. One enzyme activity unit represents the amount of MN_x that catalyzes the decomposition of 1 μmol H₂O₂ within 1 minute. **c**, Comparison of the stability of CAT-like activities between MN_x and natural enzymes. All tests were performed at room temperature. **d**, The energy barriers of MN_x in the CAT-mimic reaction simulated by DFT. **e-f**, Schematic illustration of the cyclable catalytic CAT processes for **(e) Fe-Cu-N₆** and **(f) Rh/VN₄**. **g**, The CAT processes with MN_x (Rh, V, and Fe-Cu). Dark grey, blue, red, cyan, light grey, orange, and brown balls represent H, C, N, O, Rh, V, Fe, and Cu atoms. The MN_x catalyst is represented by an asterisk (*) for clarity.

9. The inflammatory response at the wound site has the effect of preventing microbial infection. Has the author considered whether MN_x may increase the possibility of wound infection after being anti-inflammatory and antioxidant?

Reply: Thank you very much for this valuable recommendation. Before experiments, all sutures modified with MN_x were high-temperature sterilized and tools were sterile. All animals were fed and experiments were conducted under the specific pathogen-free (SPF) environment. After suturing on the scalp, wounds were disinfected with iodophors. Per your suggestions, we collected secretions from the surface of the scalp and cultured them on the day 3 post injury, aiming to detect whether exists microbial infection at the wound site. The results demonstrated that there was only Staphylococcus xylose in the culture in all groups (**Figure R3**), showing weak pathogenicity to the wound, and there were no other bacteria, which can lead to wound infection like Staphylococcus aureus, Pseudomonas aeruginosa and Escherichia coli (*Int. J Pharmaceut.* 2013, 441, 181-191; *Mater. Sci. Eng. C.* 2021, 129, 112422). Therefore, it is rational to conclude that MN_x-based sutures do not induce microbial infection at the wound site under the standard and strict disinfection operation during anti-inflammatory and antioxidant process.

Referred to your suggestion, we have added corresponding descriptions about the experiment details in the revised manuscript.

Added in the revised manuscript (Lines 22-23 and 1, Pages 32-23):

“Notably, the MN_x-sutures were treated with a high-temperature sterilization procedure before use, and the entire animal experiments were conducted under a specific pathogen-free (SPF) environment”

Figure R3. Bacterial growth at the wound site of scalp on day 3 post-injury. Only *Staphylococcus xylose* was detected in each group.

Comments by Reviewer #2

In this manuscript, through a unique bilateral reaction mechanism that decreases reaction energy barriers, Zhang et al constructed a series of single-atom artificial enzymes RhN₄, VN₄, and Fe-Cu-N₆, showing potent catalytic activities beyond natural enzymes. The RhN₄, VN₄, and Fe-Cu-N₆ displayed high peroxidase-like (POD-like) activity, catalase-like (CAT-like) activity, superoxide dismutase-like (SOD-like) activity, and glutathione peroxidase-like (GPx-like) activity. Moreover, surgical sutures based on RhN₄ and VN₄ showed recyclable catalytic features without apparent decay in 1 month and accelerated scalp healing from brain trauma via promoting the vascular endothelial growth factor (VEGF) and diminishing oxidative stress and inflammation.

The authors presented a novel yet cost-effective approach to construct a kind of efficient and stable artificial enzyme and successfully used RhN₄, VN₄, and Fe-Cu-N₆ in the field of healing traumatic brain injury. The overall study was well organized and performed and the goals of this research were of particular interest in the field of nano-catalysis and bioengineering. The specific issues are suggested to be addressed as following:

1. The immune cells like macrophages, Tregs, natural killer cell, B cells as well as T cells play an important role in wound repairing. So, I strongly suggest the authors to analyze the changes of immune cells after various treatment via flow cytometry.

Reply: Thank you very much for this valuable recommendation. Per your suggestions, we evaluated the percentage of M1 and M2 macrophages around the wounds in the scalp on day 6 post injury after different sutures by flow cytometry. As shown in **Figure 6f, g** and **Figure S44**, M1 phenotype macrophages decreased and M2 phenotype macrophages increased, further confirming that modified sutures could speed up the wound closure. In addition, we analyzed the changes of immune cells like Tregs, natural killer cell, B cells as well as T cells after various treatment in the blood via flow cytometry. As shown in **Figure S45**, mice could induce systemic immune response after brain injury and gradually recovered to the normal with MN_x-based sutures. The corresponding descriptions and figure have also been added in the revised manuscript.

Added in the revised manuscript (Lines 22-23 and 1-7, Pages 14-15):

“As shown in **Figures 6f, g**, and **Figure S44**, the population of M1 macrophages decreased and was polarized into the M2 phenotype, further confirming the anti-inflammatory effects of MN_x sutures. In addition, changes in other immune cells like natural killer cells, B cells, and T cells after various treatments in whole blood were also analyzed by flow cytometry. As shown in **Figure S45**, mice could induce systemic immune responses after brain injuries and gradually recovered to normal with MN_x-based sutures. Besides, the tissue macrophage marker CD68 was upregulated after scalp injuries but recovered by MN_x sutures, while the vascular epidermal growth factor (VEGF) was also ameliorated by MN_x sutures (**Figures 6h-j**, **S43d**, and **S46**).”

Added in the revised manuscript (Lines 18-24 and 1-5, Pages 34-35):

“Flow cytometry

To evaluate the changes of immune cells in the wound and blood, we collected the scalp and blood in different groups on day 6 post brain injuries. The fresh wound tissues were digested with collagenase and hyaluronidase (Stemcell, 07919) overnight at 37 °C to obtain single cells for flow cytometry analysis (BD FACSCanto II). Lymphocytes in the blood were collected with Ficoll kits (Solarbio, P8620). Lymphocytes were stained with PE anti-mouse F4/80 (Biolegend, 123110), APC anti-mouse CD80 (Biolegend, 104714), FITC anti-mouse CD206 (MMR) (Biolegend, 141704), APC/Cyanine7 anti-mouse CD3 (Biolegend, 100222), PE/Cyanine7 anti-mouse CD8a (Biolegend, 100722), FITC anti-mouse CD4 (Biolegend, 100406), PE anti-mouse NK-1.1 (Biolegend, 156504), PerCp anti-mouse CD45 (Biolegend, 103130), APC anti-mouse CD19 (Biolegend, 152410), APC anti-mouse CD25 (Biolegend, 101909), and PE-Foxp3 (FJK-16s) (Thermo, 12-5773-82) antibodies.

”

Changed in the revised manuscript (Figure 6, Pages 47-48):

Figure 6. The scalp healing process with MN_x sutures. **a**, Schematic illustration of the scalp closure process modulated by MN_x sutures. **b-c**, Representative residual wound (**b**) and photographs of scalp healing (**c**) over time with and without MN_x sutures. **d**, Indicators for oxidative stress, including GSH, MDA and SOD, of the scalp with or without MN_x sutures on day 3 post-injury in RhN₄, VN₄, and Fe-Cu-N₆ groups (n=3 per group). **e**, Quantitative ELISA analysis of IL-6, IL-1β and TNF-α levels in scalp tissues with or without MN_x sutures on day 3 post-injury in RhN₄, VN₄ and Fe-Cu-N₆ groups (n= 3 per group), respectively. **f-g**, The flow cytometry analysis of (**f**) M1 and (**g**) M2 macrophages in the scalp after RhN₄, VN₄ and Fe-Cu-N₆ treatments. **h**, Quantitative ELISA analysis of VEGF levels in scalp tissues with or without MN_x sutures on day 3 post-injury. **i-j**, Immunofluorescence staining of (**j**) CD68 and (**i**) the quantitative analysis in the injured scalp on day 3 post wound injury in RhN₄, VN₄, and Fe-Cu-N₆ groups (n= 4 per group). **k**, Immunohistochemistry of CD31 and (**i**) the quantitative analysis in scalp tissues on day 4 post wound injury in RhN₄, VN₄, and Fe-Cu-N₆ groups (n= 4 per group). TBI groups were sutured with pristine PGA sutures for FPI. The control groups were only depilated as wild animals. Data were presented as mean ± SEM and compared with the TBI group by one-way ANOVA with the one-sided Tukey's multiple comparisons test. **p* < 0.05, ***p* < 0.01, ****p* < 0.001 and *****p* < 0.0001 versus the TBI group.

Added in the revised supporting information (Figure S44 and S45, Pages 45-46):

Figure S44. The flow cytometry analysis of (a) M1 and (b) M2 macrophages in the scalp after FeN₄ and CuN₄ treatments.

Figure S45. The flow cytometry analysis of (a) B cells, (b) NK cells, (c) Tregs, and (d, e) T cells in the blood after various treatments.

2. I have concerned that the metal on the MN_x-based sutures could spread and release into the blood, which might lead to poisoning. I strongly suggest authors to detect the concentration of metal in the blood after various treatment.

Reply: Thank you very much for the useful comments. The metal content in the MN_x single-atom enzymes is low. After making MN_x-based sutures, the amount of metal elements loaded by the sutures was detected by ICP-MS with a mass ratio of about 0.01 - 0.02%. When the wounds were treated with MN_x-based sutures, small amounts of MN_x enzymes could spread and release into the blood. According to your suggestion, we detected the metal content in the blood in the normal and the MN_x-treated groups on the day 6 post injury, respectively. The results showed that the metal levels in the blood were comparable to those of normal mice. This may be attributed to the metabolism of the mice on the one hand and to the fact that the metal levels are below the detection limit of the instrument on the other hand. We have also added the figure of metal content and related discussions in the revised manuscript.

Added in the revised manuscript (lines 22 and 1-2, Pages 15-16):

“In addition, no metal elements were detected in the blood of MN_x-treated mice on day 6 after injuries, suggesting that the MN_x-based sutures are biologically safe (Figure S54).”

Added in the revised supporting information (Figure S54, Page 55):

Figure S54. Metal contents in the blood in the MN_x-treated group on day 6 post-injury. The baseline represents the corresponding metal content in the blood of normal mice.

3. *It would be better to add important quantitative data in the abstract to highlight the significance of this manuscript.*

Reply: Thanks for your careful review and grateful comments. Based on your suggestion, we have rewritten the abstract to highlight the innovative and importance of this work.

Changed in the revised manuscript (Page 2):

“**ABSTRACT.** Regenerable artificial enzyme mimetics with high catalytic stability and sustainability are promising as a substitute for naturally-occurring enzymes to some extent. However, they are usually limited by insufficient catalytic activities and non-specific selectivity. Herein, we developed single-atom artificial enzymes of RhN₄, VN₄, and Fe-Cu-N₆ with extraordinary catalytic activities surpassing natural enzymes. Notably, Rh/VN₄ preferably formed an Rh/V-O-N₄ active center with circular catalytic processes via decreasing reaction energy barriers by 0.2 eV and showed 4 and 5-fold higher affinities in the peroxidase-like (POD-like) activity than FeN₄ and the natural horseradish peroxidase, respectively. Furthermore, RhN₄ presented an over 20-fold improved affinity in the catalase-like (CAT-like) activity compared to the natural CAT; Fe-Cu-N₆ displayed high selectivity towards the superoxide dismutase-like (SOD-like) activity; VN₄ favored a 7-fold higher glutathione peroxidase-like (GPx-like) activity than the natural GPx. The Rh/VN₄ mediated a “two-sided oxygen-linked” catalytic reaction path, resulting in high-performance biocatalytic activities in a mechanism distinct from the previously-reported single-atom catalysts. Bioactive sutures developed using RhN₄ and VN₄ showed recyclable catalytic features without apparent decay in one month and accelerated scalp healing from brain trauma by promoting the vascular endothelial growth factor (VEGF), regulating the immune cells like macrophages, and diminishing oxidative stress and inflammation.”

Minor issues:

1) *In Page 5, Line 99 in the main text, “(Figure 1j)” should be “(Figure 2j)”.*

Reply: Thank you very much for the careful review. We have carefully checked and corrected the writing errors throughout the revised manuscript.

Changed in the revised manuscript (Lines 18-20, Page 5):

“By analyzing the XANES of RhN₄ (Figure 2j), we can observe that the Rh species become more positively charged than Rh(acac)₃, which may originate from the Rh-N₄ unit.”

2) *Some references about wound repairing such as Matter, 2021,4 (9), 2985-3000, Small Structures, 2021, 2(7), 2100002 are suggested to be added.*

Reply: Thank you very much for this kind suggestion. These references you mentioned really strengthen the persuasiveness of our article and are very helpful for the discussion of wound repairing. In view of your suggestions, we have cited these references in the corresponding manuscripts.

Added in the references of the revised manuscript (Page 38):

- 55 Hu, J. *et al.* Mechanically active adhesive and immune regulative dressings for wound closure. *Matter* **4**, 2985-3000 (2021).
- 57 Zhu, J. *et al.* Reactive oxygen species scavenging sutures for enhanced wound sealing and repair. *Small Structures* **2**, 2100002 (2021).

REVIEWER COMMENTS

Reviewer #1 (Remarks to the Author):

The authors have performed substantially experiments that addressed most of major concerns raised by reviewer. However, some minor concerns should be addressed before the acceptance.

1. Figure 3 and 4 must be wrong! They are exactly same figures. Please carefully check them.

2. About the title and nomenclature in this work. All the nanomaterials are typically single atom nanozymes. The reviewer suggest the authors should specifically rename all the materials to single atom/ dual atom nanozymes, but not artificial enzymes.

3. Some beneficial discussions should be added for the latest advances of nanozyme, and following references may be helpful, e.g. Langmuir, 2022, 38 (12), 3617-3622; ACS Nano, 2021, 15 (10), 15645-15655; Nano Today, 2021, 40, 101269.

Reviewer #2 (Remarks to the Author):

I think that the authors have satisfactorily addressed the comments provided with the original version, and I believe the paper is suitable for publication.

Our Replies to the Reviewer' Comments and the Revisions Made to the Manuscript

Comments by Reviewer #1

The authors have performed substantially experiments that addressed most of major concerns raised by reviewer. However, some minor concerns should be addressed before the acceptance.

1. Figure 3 and 4 must be wrong! They are exactly same figures. Please carefully check them.

Reply: Thank you very much for the careful review. The correct figure 3 have been added in the revised manuscript. Meanwhile, we have carefully checked the figures throughout the revised manuscript.

Changed in the revised manuscript (Fig. 3, Page 45):

2. About the title and nomenclature in this work. All the nanomaterials are typically single atom nanozymes. The reviewer suggest the authors should specifically rename all the materials to single atom/ dual atom nanozymes, but not artificial enzymes.

Reply: Thank you very much for your valuable recommendation. Based on your suggestions, We have renamed the title of this work and have changed the artificial enzymes to single-atom nanozymes in the corresponding revised manuscript and figures, as well as title.

For example,

Changed in the revised manuscript of title (Lines 1-2, Page 1):

“Single-Atom **Nanozymes** Catalytically Surpassing Naturally-occurring Enzymes as Sustained Stitching for Brain Trauma”

3. *Some beneficial discussions should be added for the latest advances of nanozyme, and following references may be helpful, e.g. Langmuir, 2022, 38 (12), 3617-3622; ACS Nano, 2021, 15 (10), 15645-15655; Nano Today, 2021, 40, 101269.*

Reply: Thank you very much for the useful suggestion. These references you mentioned really strengthen the persuasiveness of our article and are very helpful for the discussion of latest advances of nanozyme. In view of your suggestions, we have added the related discussions and cited these references in the corresponding manuscripts.

Added in the revised manuscript (Lines 6-8, Page 3):

“As a fast growing category of artificial enzymes, the emerging nanozyme can actively tailor biocatalytic activities and selectivity via its flexible atomic structures and molecular engineering.²³⁻²⁶”

Added in the references of the revised manuscript (Page 37):

- 23 Zandieh, M. & Liu, J. Surface science of nanozymes and defining a nanozyme unit. *Langmuir* **38**, 3617-3622 (2022).
- 24 Zandieh, M. & Liu, J. Nanozyme catalytic turnover and self-limited reactions. *ACS Nano* **15**, 15645-15655 (2021).
- 25 Wei, H. *et al.* Nanozymes: A clear definition with fuzzy edges. *Nano Today* **40**, 101269 (2021).
- 26 Tang, G., He, J., Liu, J., Yan, X. & Fan, K. Nanozyme for tumor therapy: Surface modification matters. *Exploration* **1**, 75-89 (2021).

REVIEWERS' COMMENTS

Reviewer #1 (Remarks to the Author):

The authors have fully addressed my concerns. I think it can be accepted now.